# Adapting Precomputed Features for Efficient Graph Condensation

**Yuan Li** [1]  **Jun Hu** [1]  **Zemin Liu** [2]  **Bryan Hooi** [1]  **Jia Chen** [3]  **Bingsheng He** [1]

## Abstract

Graph Neural Networks (GNNs) face significant computational challenges when handling large-scale graphs. To address this, Graph Condensation (GC) methods aim to compress large graphs into smaller, synthetic ones that are more manageable for GNN training. Recently, trajectory matching methods have shown state-of-the-art (SOTA) performance for GC, aligning the model's training behavior on a condensed graph with that on the original graph by guiding the trajectory of model parameters. However, these approaches require repetitive GNN retraining during condensation, making them computationally expensive. To address the efficiency issue, we completely bypass trajectory matching and propose a novel two-stage framework. The first stage, a precomputation stage, performs one-time message passing to extract structural and semantic information from the original graph. The second stage, a diversity-aware adaptation stage, performs class-wise alignment while maximizing the diversity of synthetic features. Remarkably, even with just the precomputation stage, which takes only seconds, our method either matches or surpasses 5 out of 9 baseline results. Extensive experiments show that our approach achieves comparable or better performance while being $96\times$ to $2,455\times$ faster than SOTA methods, making it more practical for large-scale GNN applications. [1]

## 1. Introduction

**Graph Condensation.** Graph learning through GNNs (Kipf & Welling, 2016; Hamilton et al., 2017) has significantly advanced graph data analysis, providing insights into com-

plex graph structures such as social networks (Fan et al., 2019; Zhang et al., 2022) and molecular structures (Guo et al., 2021; Gasteiger et al., 2021). However, real-world large-scale graphs pose significant challenges for training GNNs (Huang et al., 2021; Gao et al., 2024). To address this, graph condensation techniques (Jin et al., 2022b;a) propose generating a condensed graph from a large original graph, enabling models trained on the condensed graph to be directly applied to the original graph, achieving comparable performance on both graphs. GC enhances training efficiency by reducing computational costs on large-scale graphs. Recent studies further demonstrate that GC facilitates efficient GNN training with minor performance loss (Gao et al., 2024; Zhang et al., 2024).

**Prior Work.** GC methods encompass a variety of approaches, including coreset selection (Welling, 2009; Sener & Savarese, 2018), gradient matching (Jin et al., 2022b;a; Yang et al., 2024; Fang et al., 2024), distribution matching (Liu et al., 2022; Xiao et al., 2024; Gao et al., 2025), and trajectory matching (Zheng et al., 2024; Zhang et al., 2024). Among them, **trajectory matching** has emerged as the SOTA method due to its ability to align the training behavior effectively. These methods collect training trajectories from the original graph to guide the training process on the condensed graph, ensuring consistent training behavior between the two graphs. Notably, existing trajectory-based methods operate in a Structure-Free (SF) manner, producing condensed graphs without edges, which simplifies the training process. Although these condensed graphs lack edges, they support GNN node classification tasks by adding self-loop edges for each node, ensuring compatibility with GNNs and achieving SOTA.

**Efficiency Issues with SOTA Method.** Despite the leading performance, trajectory-based methods require a substantial number of trajectories to achieve optimal results. As illustrated in Figure 1, trajectory collection involves repeatedly training the model from scratch (e.g., 200 times), taking a remarkably long time (e.g., 452 hours on million-node graphs) and dominating the overall condensation time. This inefficiency hinders their applications at scale, e.g., on social networks or e-commerce graphs.

To address the inefficiencies, we propose a novel Graph Condensation framework via a Precompute-then-Adapt ap-

[1]National University of Singapore [2]Zhejiang University [3]GrabTaxi Holdings Pte. Ltd.. Correspondence to: Jun Hu <jun.hu@nus.edu.sg>.

*Proceedings of the $42^{nd}$ International Conference on Machine Learning*, Vancouver, Canada. PMLR 267, 2025. Copyright 2025 by the author(s).

[1]Our code and data are available at https://github.com/Xtra-Computing/GCPA.

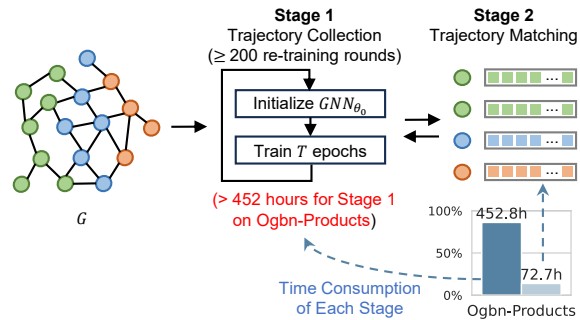

Figure 1: Trajectory-based methods require repetitive GNN re-training during trajectory collection, which can be highly time-consuming. This stage accounts for the majority of the total runtime in such methods.

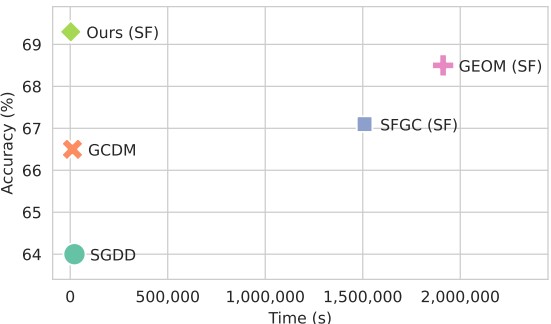

Figure 2: Performance *vs.* condensation time on Ogbn-Products. Our framework significantly reduces condensation time, bypassing the costly trajectory collection stage and outperforming SOTA methods. Here, SF denotes structure-free graph condensation, where the condensed graphs possess no edges.

proach (GCPA). Our method first employs a **precomputation stage** and then a **diversity-aware adaptation stage**, eliminating the need for repetitive re-training with different random initializations. The precomputation stage extracts structural and semantic information from the original graph, achieving competitive performance in a short time. The adaptation stage further refines the precomputed features through class-wise alignment and diversity maximization. As a result, we achieve competitive performance (-1.5% to +2.4%) on node classification tasks with substantially faster training time (96× to 2,455×) compared to SOTA (GEOM (Zhang et al., 2024)), as exemplified in Figure 2.

We summarize the key contributions of our work as follows:

- We propose GCPA, an **efficient** graph condensation framework with a precomputation and an adaptation stage. Compared to SOTA methods, our framework avoids the costly repetitive re-training during trajectory collection, achieving significant efficiency improvements.

- Our framework is **effective**. The precomputation stage, which extracts structural and semantic information from the original graph, already matches or surpasses 5 out of 9 baselines. With the adaptation stage, we further enhance performance with class-wise feature alignment and diversity maximization, achieving SOTA performance on 4 out of 7 datasets.

- Through extensive experiments on benchmark datasets, we demonstrate that our method achieves better or comparable accuracy while being up to 2,455× faster than existing methods.

## 2. Preliminaries

Let $\mathcal{G} = \{\mathbf{X}, \mathbf{A}, \mathbf{Y}\}$ denote a graph, where $\mathbf{X} \in \mathbb{R}^{N \times d}$ denotes the node feature matrix with $N$ nodes and $d$-

dimensional features, $\mathbf{A} \in \{0, 1\}^{N \times N}$ represents the adjacency matrix, $\mathbf{Y} \in \mathbb{R}^{N \times C}$ denotes the ground-truth one-hot node labels among $C$ classes, and $\mathbf{y} \in \mathbb{R}^N$ records the labels in vector form. Graph condensation aims to generate a synthetic graph (or condensed graph, used interchangeably) corresponding to an existing graph such that a model trained on the synthetic graph is effective when applied to the original graph. Formally, given an original graph $\mathcal{T} = \{\mathbf{X}, \mathbf{A}, \mathbf{Y}\}$ with $N$ nodes, GC aims to generate a smaller synthetic graph $\mathcal{S} = \{\mathbf{X}', \mathbf{A}', \mathbf{Y}'\}$ with $N'$ nodes such that a GNN trained on $\mathcal{S}$ achieves similar performance on $\mathcal{T}$ as another GNN trained directly on $\mathcal{T}$. In particular, structure-free graph condensation emerges as a storage-efficient graph condensation approach where the adjacency matrix is set to an identity matrix, $\mathbf{A}' = \mathbf{I}$, so the synthetic graph does not contain structural information.

Node classification is a prevalent task simplified by graph condensation, involving label assignment based on node features and graph structure. Formally, given a graph $\mathcal{G} = \{\mathbf{X}, \mathbf{A}\}$, and a subset of nodes $N_L \subseteq N$ with known labels $\mathbf{Y}_L \in \mathbb{R}^{N_L \times C}$, the transductive semi-supervised node classification task involves predicting labels $\mathbf{Y}_U \in \mathbb{R}^{N_U \times C}$ for an unlabeled subset of nodes $N_U \subseteq N$. The optimization goal can be formulated as a bi-level problem:

$$\min_{\mathcal{S}} \mathcal{L}(\text{GNN}_{\boldsymbol{\theta}_\mathcal{S}}(\mathbf{X}, \mathbf{A}), \mathbf{Y})$$
$$\text{s.t.} \quad \boldsymbol{\theta}_\mathcal{S} = \arg\min_{\boldsymbol{\theta}} \mathcal{L}(\text{GNN}_{\boldsymbol{\theta}}(\mathbf{X}', \mathbf{A}'), \mathbf{Y}') \quad (1)$$

where $\boldsymbol{\theta}$ denotes the learnable parameters of an $L$-layer GNN model, $\boldsymbol{\theta}_\mathcal{S}$ represents the optimal GNN parameters learned on the synthetic graph, and $\mathcal{L}$ is a loss function evaluating the node classification performance. Existing graph condensation approaches optimize this bi-level problem to learn an optimal synthetic graph $\mathcal{S}$ such that a trained GNN

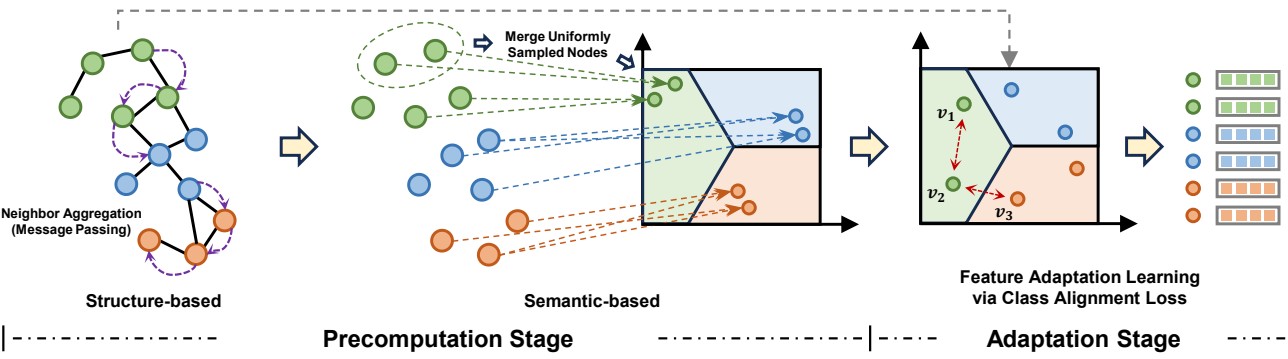

Figure 3: Overall pipeline of the proposed GCPA condensation framework. (1) Structure-based precomputation: Neighbor aggregation is performed to capture structural dependencies. (2) Semantic-based precomputation: Nodes are grouped by semantic relevance and uniformly sampled. (3) Adaptation learning: Synthetic features v1 and v2 are pushed away through diversity constraints, while v2 and v3 are pushed away through sampled negative pairs.

with parameters $\boldsymbol{\theta}_{\mathcal{S}}$ yields optimal performance on $\mathcal{T}$. However, the bi-level optimization problem is computationally intensive, as it involves nested optimization loops. To mitigate this issue, we introduce our framework that directly optimizes synthetic node features for improved efficiency.

## 3. GCPA Framework

The overview of the Graph Condensation framework via a Precompute-then-Adapt approach (GCPA) is shown in Figure 3. We introduce two stages—a precomputation stage and a diversity-aware adaptation stage—to produce structure-free synthetic data. The precomputation stage involves structure-based neighbor aggregation and semantic-based aggregation on the original graph, achieving competitive performance in a relatively short time. The adaptation learning stage further refines the precomputed features using a class-wise feature alignment objective to improve performance with minor additional costs.

### 3.1. Structure-based Precomputation

In the context of graph-based learning models, neighbor information aggregation refers to the process by which node features are enriched with structural information from neighboring nodes. This process allows a node's feature vector to incorporate not just its own information but also that of its surrounding neighborhood. Such aggregation is critical for capturing relationships and dependencies in graph-structured data.

Drawing inspiration from the graph diffusion process (Gasteiger et al., 2019; Frasca et al., 2020; Hu et al., 2024b;a), we leverage neighbor structural information to pre-process the original node features. The goal of graph diffusion is to smooth node features based on the underly-

ing graph's topology, facilitating the effective propagation of information across nodes. The structure-based precomputed features $\mathbf{H}$ with $K$-hop neighbor aggregation can be recursively computed as:

$$\mathbf{H}^{(k)} = (1 - \alpha)\hat{\mathbf{A}}\mathbf{H}^{(k-1)} + \alpha\mathbf{H}^{(0)} \quad \text{for } k = 1, 2, \ldots, K$$

$$\text{with} \quad \hat{\mathbf{A}} = \tilde{\mathbf{D}}^{-\frac{1}{2}}\tilde{\mathbf{A}}\tilde{\mathbf{D}}^{-\frac{1}{2}}, \quad \tilde{\mathbf{A}} = \mathbf{A} + \mathbf{I}_N \tag{2}$$

where $\mathbf{H}^{(0)} = \mathbf{X}$ represents the node feature matrix, $K$ denotes the number of aggregation hops, and $\mathbf{H} = \mathbf{H}^{(K)}$ is the output of the last layer. $\tilde{\mathbf{D}}$ is the degree matrix of $\tilde{\mathbf{A}}$, where $\tilde{\mathbf{D}}_{ii} = \sum_j \tilde{\mathbf{A}}_{ij}$. The coefficient $\alpha$ controls the contribution of raw features to each hop. Having processed the structural information, we omit the edges in the follow-up semantic-based precomputation as shown in Figure 3, focusing on processing semantic information.

### 3.2. Semantic-based Precomputation

To condense a set of $N$ aggregated features into $N'$ synthetic node features, we perform semantic-based precomputation by merging uniformly sampled original nodes within each class. Specifically, for each synthetic node $v_i$ with class label $c \in \{1, 2, \ldots, C\}$, we uniformly sample a subset of original nodes in the same class. Then, we compute the semantic-based features by taking the mean of the aggregated features of the sampled nodes:

$$\hat{\mathbf{X}}'_i = \frac{1}{M} \sum_{j \in \mathcal{S}_i} \mathbf{H}_j \quad \text{for } i = 1, 2, \ldots, N' \tag{3}$$

$$\text{with} \quad \mathcal{S}_i \subseteq \mathcal{I}_{\mathbf{y}_i}, \quad |\mathcal{S}_i| = M, \quad \mathcal{I}_c = \{i \mid \mathbf{y}_i = c\}$$

where $\mathcal{S}_i$ is the set of sampled original nodes for synthetic node $i$, $\mathcal{I}_c$ denotes the indices of original nodes belonging to class $c$, $M$ is the number of sampled nodes for each synthetic node.

This semantic-based precomputation process effectively condenses the semantic information of multiple nodes within the same class into a single synthetic node. Furthermore, by maintaining the class distribution in $\mathbf{Y}$ through proportional sampling, we fix the synthetic labels $\mathbf{Y}'$ to preserve the original class proportions. Consequently, we obtain the precomputed condensed dataset $\{\hat{\mathbf{X}}', \mathbf{Y}'\}$, where $\hat{\mathbf{X}}' \in \mathbb{R}^{N' \times d}$ and $\mathbf{Y}' \in \mathbb{R}^{N' \times C}$.

### 3.3. Feature Adaptation Learning

Given the limited number of condensed nodes, it is crucial that these nodes ideally depict the overall representations of their respective classes (depicted by the background color in Figure 3). Our precomputation stage effectively captures the structural and semantic features of the original graph, setting up a solid foundation for downstream learning. Since the precomputation stage is not directly optimized for the final objective, we further integrate an adaptation learning stage that adjusts the class-wise representations.

To achieve better class-wise representation separation, we consider a contrastive loss to enhance node features for classification utility (Joshi et al., 2022). For the task of graph condensation, we propose to align the condensed features with the original precomputed features using a class-wise adaptation loss. Specifically, we introduce an adaptation module $f_{\text{adapt}} : \mathbb{R}^{N' \times d} \to \mathbb{R}^{N' \times d}$, implemented as a Multi-Layer Perceptron (MLP) (Rosenblatt, 1958) with $F$ layers, to better depict the overall representations:

$$\mathbf{Z}' = \beta \hat{\mathbf{X}}' + (1 - \beta) f_{\text{adapt}}(\hat{\mathbf{X}}') \qquad (4)$$

where $\beta$ is a hyperparameter controlling the contribution of precomputed representations, $\mathbf{Z}'$ represents the adapted synthetic representations. We adopt $\{\mathbf{Z}', \mathbf{Y}'\}$ as the final condensed dataset after the learning process. Notably, $\mathbf{X}'$ is treated as a fixed, non-trainable condition during training, thereby imposing a permanent constraint on $\mathbf{Z}'$. This contrasts with prior approaches (Jin et al., 2022b; Zheng et al., 2024; Zhang et al., 2024), which use randomly sampled features only for initialization and allow these signals to drift away throughout optimization.

We further construct the contrastive samples by first sampling a sufficient number of anchor nodes $\mathbf{H_i}$ from the precomputed representations $\mathbf{H}$. These anchors serve as learning targets during the adaptation stage, encouraging the synthetic representations to preserve the original feature distributions. The first introduced constraint is class-wise alignment, where, for each anchor node in the original graph, we sample a synthetic node belonging to the same class as a positive sample and a set of arbitrary synthetic nodes from different classes as negative samples. Additionally, we introduce an intra-class diversity constraint to encourage synthetic features within each class to be dissimilar.

With the sampled pairs, we optimize an adaptation loss as follows:

$$
\begin{aligned}
\mathcal{L} = -\,\mathbb{E}_{\{i,j|\mathbf{y}_i = \mathbf{y}'_j\}} \Bigg( & \log \sigma\Big(\langle \mathbf{H}_i, \mathbf{Z}'_j \rangle\Big) \\
& + \underbrace{\mathbb{E}_{t \sim \text{Uniform}\{\,t:\,\mathbf{y}'_t \neq \mathbf{y}_i\,\}} \log \sigma\Big(-\langle \mathbf{H}_i, \mathbf{Z}'_t \rangle\Big)}_{\text{class-wise alignment}} \\
& + \gamma\, \underbrace{\mathbb{E}_{t \sim \text{Uniform}\{\,t:\,\mathbf{y}'_t = \mathbf{y}_i\,\}} \log \sigma\Big(-\langle \mathbf{H}_i, \mathbf{Z}'_t \rangle\Big)}_{\text{intra-class diversity}} \Bigg)
\end{aligned}
\qquad (5)
$$

where $\langle \mathbf{H}_i, \mathbf{Z}'_j \rangle$ denotes the inner product between the $i$-th anchor node's representation $\mathbf{H}_i$ and the $j$-th synthetic node's adapted representation $\mathbf{Z}'_j$, $t$ is the index of a random negative sample from the synthetic dataset, and $\sigma(x) = 1/(1 + \exp(-x))$ is the sigmoid function. The adaptation module refines the precomputed representations to better align synthetic and original node representations, enhancing the generalization of the condensed features. We optimize the model using the AdamW optimizer (Loshchilov & Hutter, 2017), with learning rate $\eta$, weight decay $\lambda$, and betas $(\beta_1, \beta_2)$ as hyperparameters.

### 3.4. Complexity Analysis

We analyze the time complexity using GCN as the backbone for GC. For GCPA, the time complexity is $O(KEd + PF(N'd + N'd^2))$, where $E$ is the number of edges of the original graph and $P$ is the number of epochs. The first term corresponds to precomputation, while the second accounts for adaptation. In contrast, the time complexity of SOTA (GEOM) is $O(TPL(Ed + Nd^2 + N'd + N'd^2))$, where $T$ represents the number of repetitive GNN retraining iterations for trajectory matching. Compared to GCPA, GEOM includes an additional multiplicative factor $T$, which can be large (e.g., in the hundreds) and significantly increases GEOM's time complexity. Conversely, GCPA bypasses trajectory matching, eliminating the dependence on $T$ and making it substantially more efficient than GEOM.

## 4. Experiments

In this section, we conduct experiments to validate the effectiveness and efficiency of the proposed framework.

### 4.1. Experimental Setup

**Datasets.** Following GCondenser (Liu et al., 2024), a comprehensive graph condensation benchmark, our experiments are conducted on seven benchmark datasets including three smaller networks: CiteSeer, Cora, and PubMed (Kipf & Welling, 2016), and four larger graphs: Ogbn-arxiv, Ogbn-products (Hu et al., 2020), Flickr (Zeng et al., 2020), and Reddit (Hamilton et al., 2017). We use the public data splits

Table 1: Summary of dataset statistics.

| Setting | Dataset | # Train/Val/Test Nodes | # Nodes | # Edges | # Features | # Classes |
|---------|---------|------------------------|---------|---------|------------|-----------|
| Transductive | CiteSeer | 120/500/1,000 | 3,327 | 4,732 | 3,703 | 6 |
| | Cora | 140/500/1,000 | 2,708 | 5,429 | 1,433 | 7 |
| | PubMed | 60/500/1,000 | 19,717 | 88,648 | 500 | 3 |
| | Ogbn-arxiv | 90,941/29,799/48,603 | 169,343 | 1,166,243 | 128 | 40 |
| | Ogbn-products | 196,615/39,323/2,213,091 | 2,449,029 | 61,859,140 | 100 | 47 |
| Inductive | Flickr | 44,625/22,312/22,313 | 89,250 | 899,756 | 500 | 7 |
| | Reddit | 153,431/23,831/55,703 | 232,965 | 57,307,946 | 602 | 41 |

for fair comparisons. The dataset statistics and settings are detailed in Table 1. For CiteSeer, Cora, and PubMed datasets, row feature normalization is applied to prepare the data. For Ogbn-arxiv, Flickr, and Reddit datasets, we apply feature standardization. The Ogbn-products dataset retains its feature processing as defined by OGB (Hu et al., 2020).

**Baselines.** We compare our proposed framework to the baselines in the following categories: (i) Coreset approach: K-Center (Sener & Savarese, 2018). (ii) Gradient matching approaches: GCond (Jin et al., 2022b), SGDD (Yang et al., 2024), and EXGC (Fang et al., 2024). (iii) Distribution matching approaches: GCDM (Liu et al., 2022), SimGC (Xiao et al., 2024), and CGC (Gao et al., 2025). (iv) Trajectory matching approaches: SFGC (Zheng et al., 2024) and GEOM (Zhang et al., 2024). We report results based on the following sources, depending on their availability: (a) official results reported in the original papers of each baseline, (b) reproduced results provided by the GCondenser benchmark (Liu et al., 2024), and (c) results obtained by running the official code.

**Backbone Models.** We use GCN (Kipf & Welling, 2016) and SGC (Wu et al., 2019) as backbone models during condensation and evaluation for fair comparisons. In the cross-architecture evaluation, we use more models including MLP (Rosenblatt, 1958), GAT (Veličković et al., 2018), ChebNet (Defferrard et al., 2016), GraphSAGE (Hamilton et al., 2017), and APPNP (Gasteiger et al., 2018).

The details of the evaluation schemes and hyperparameter settings are provided in Appendix A and B, respectively.

### 4.2. Performance Comparison

We present the performance of different graph condensation approaches using the GCN backbone in Table 2. Additionally, the performance of these approaches with the SGC backbone is shown in Table 7, located in the Appendix. Based on these results, we make the following observations:

- The coreset approach, K-Center, which typically employs conventional machine learning techniques, fails to provide good condensation results on all datasets. This highlights the non-trivial nature of graph condensation tasks, which necessitate substantial effort.

- Two distinct categories of graph condensation methods, including gradient matching and distribution matching, have both shown fair performance on different datasets. Notably, neither category consistently outperforms the other. This variation in performance suggests that multiple frameworks might be applicable to the task of graph condensation, without a universally superior approach.

- Recent advancements in trajectory matching, especially the SFGC and GEOM approaches, have demonstrated superior performance on most datasets, affirming the efficacy of trajectory-based methods. Notably, both SFGC and GEOM employ structure-free condensation, indicating that for node classification tasks, providing edges in condensed graphs may not always be necessary.

- Our proposed framework achieves superior performance on 4 out of 7 datasets, underscoring the effectiveness of our precompute-then-adapt approach.

### 4.3. Efficiency Comparison

We present a comprehensive efficiency comparison of different methods using the GCN backbone in Table 3 and the SGC backbone in Table 8 (located in the Appendix). Additionally, Figure 4 illustrates a joint analysis of both accuracy and efficiency. Based on the results, we make the following observations on the efficiency of different approaches:

- As depicted in Figure 4, GEOM, a trajectory-based method, exhibits leading performance but suffers from poor efficiency. The primary efficiency bottleneck lies in the need for repetitive re-training, which, while effective, leads to severe efficiency issues.

- As shown in Figure 4, our framework achieves comparable or leading performance across datasets. Notably, the framework is significantly more efficient than trajectory-based methods, achieving speedups ranging from $96\times$ to $2,455\times$ compared to these approaches.

Table 2: Node classification performance comparison using GCN backbone (mean±std). The best and second-best results are marked in bold and underlined, respectively. *Ours (Pre.)* is a precomputation-only variant. The *Whole* column represents the performance obtained by training on the whole dataset.

| Dataset | Ratio | K-Cen. | GCond | SGDD | GCDM | SimGC | EXGC | CGC | SFGC | GEOM | Ours (Pre.) | Ours | Whole |
|---|---|---|---|---|---|---|---|---|---|---|---|---|---|
| CiteSeer | 0.9% | $52.4_{\pm2.8}$ | $70.5_{\pm1.2}$ | $69.5_{\pm0.4}$ | $71.2_{\pm0.8}$ | $\underline{73.8_{\pm2.5}}$ | $69.2_{\pm2.0}$ | $72.5_{\pm0.2}$ | $71.4_{\pm0.5}$ | $73.0_{\pm0.5}$ | $72.1_{\pm0.2}$ | $\mathbf{75.4_{\pm0.4}}$ | $71.4_{\pm0.5}$ |
| | 1.8% | $64.3_{\pm1.0}$ | $70.6_{\pm0.9}$ | $70.2_{\pm0.8}$ | $71.9_{\pm0.7}$ | $72.2_{\pm0.5}$ | $70.1_{\pm0.7}$ | $72.4_{\pm0.2}$ | $72.4_{\pm0.4}$ | $\underline{74.3_{\pm0.1}}$ | $72.1_{\pm0.1}$ | $\mathbf{74.8_{\pm0.3}}$ | |
| | 3.6% | $69.1_{\pm0.1}$ | $69.8_{\pm1.4}$ | $70.3_{\pm1.7}$ | $72.3_{\pm1.3}$ | $71.1_{\pm2.8}$ | $70.6_{\pm0.9}$ | $72.0_{\pm0.5}$ | $70.6_{\pm0.7}$ | $\underline{73.3_{\pm0.4}}$ | $72.7_{\pm0.5}$ | $\mathbf{74.9_{\pm0.1}}$ | |
| Cora | 1.3% | $64.0_{\pm2.3}$ | $79.8_{\pm1.3}$ | $80.1_{\pm0.7}$ | $78.9_{\pm0.8}$ | $80.8_{\pm2.3}$ | $82.0_{\pm0.4}$ | $\mathbf{82.7_{\pm0.3}}$ | $80.1_{\pm0.4}$ | $\underline{82.5_{\pm0.4}}$ | $80.3_{\pm0.5}$ | $82.1_{\pm0.6}$ | $81.7_{\pm0.9}$ |
| | 2.6% | $73.2_{\pm1.2}$ | $80.1_{\pm0.6}$ | $80.6_{\pm0.8}$ | $79.4_{\pm0.6}$ | $80.9_{\pm2.6}$ | $81.9_{\pm1.0}$ | $82.3_{\pm1.3}$ | $81.7_{\pm0.5}$ | $\mathbf{83.6_{\pm0.3}}$ | $80.6_{\pm0.5}$ | $\underline{82.9_{\pm1.0}}$ | |
| | 5.2% | $76.7_{\pm0.1}$ | $79.3_{\pm0.3}$ | $80.4_{\pm1.6}$ | $79.9_{\pm0.2}$ | $82.1_{\pm1.3}$ | $82.3_{\pm0.9}$ | $\underline{82.5_{\pm0.6}}$ | $81.6_{\pm0.8}$ | $\mathbf{82.8_{\pm0.7}}$ | $80.8_{\pm0.3}$ | $82.3_{\pm0.7}$ | |
| PubMed | 0.08% | $72.1_{\pm0.1}$ | $67.6_{\pm0.4}$ | $76.7_{\pm1.1}$ | $75.9_{\pm0.6}$ | $74.4_{\pm0.2}$ | $77.8_{\pm0.1}$ | $77.6_{\pm0.4}$ | $78.4_{\pm0.1}$ | $\underline{80.1_{\pm0.3}}$ | $79.5_{\pm1.3}$ | $\mathbf{80.5_{\pm0.4}}$ | $79.3_{\pm0.3}$ |
| | 0.15% | $76.4_{\pm0.0}$ | $74.6_{\pm0.8}$ | $78.5_{\pm0.4}$ | $77.4_{\pm0.4}$ | $76.0_{\pm0.8}$ | $78.3_{\pm0.1}$ | $77.8_{\pm0.9}$ | $78.1_{\pm0.4}$ | $\underline{79.7_{\pm0.3}}$ | $79.7_{\pm0.3}$ | $\mathbf{80.9_{\pm0.3}}$ | |
| | 0.3% | $78.2_{\pm0.0}$ | $77.2_{\pm0.7}$ | $78.0_{\pm1.1}$ | $77.6_{\pm0.4}$ | $76.2_{\pm0.8}$ | $76.2_{\pm0.1}$ | $77.7_{\pm0.3}$ | $78.5_{\pm0.5}$ | $\underline{79.5_{\pm0.4}}$ | $79.3_{\pm0.3}$ | $\mathbf{81.7_{\pm0.4}}$ | |
| Arxiv | 0.05% | $47.2_{\pm3.0}$ | $59.2_{\pm1.1}$ | $60.8_{\pm1.3}$ | $63.3_{\pm0.3}$ | $63.6_{\pm0.8}$ | $57.6_{\pm0.6}$ | $64.1_{\pm0.4}$ | $65.5_{\pm0.7}$ | $65.5_{\pm0.6}$ | $60.5_{\pm0.9}$ | $\mathbf{67.2_{\pm0.3}}$ | $71.1_{\pm0.0}$ |
| | 0.25% | $56.8_{\pm0.8}$ | $63.2_{\pm0.3}$ | $65.8_{\pm1.2}$ | $59.6_{\pm0.4}$ | $66.4_{\pm0.3}$ | $62.3_{\pm0.3}$ | $66.4_{\pm0.1}$ | $66.1_{\pm0.4}$ | $\mathbf{68.8_{\pm0.2}}$ | $64.6_{\pm0.4}$ | $\underline{67.7_{\pm0.2}}$ | |
| | 0.5% | $60.3_{\pm0.4}$ | $64.0_{\pm1.4}$ | $66.3_{\pm0.7}$ | $62.4_{\pm0.1}$ | $66.8_{\pm0.4}$ | $65.0_{\pm0.8}$ | $67.2_{\pm0.4}$ | $66.8_{\pm0.4}$ | $\mathbf{69.6_{\pm0.2}}$ | $65.5_{\pm0.3}$ | $\underline{68.1_{\pm0.3}}$ | |
| Products | 0.025% | $55.4_{\pm0.8}$ | $63.7_{\pm0.3}$ | $64.0_{\pm0.4}$ | $66.5_{\pm0.1}$ | $63.3_{\pm1.1}$ | $62.1_{\pm0.7}$ | $68.0_{\pm0.1}$ | $67.1_{\pm0.2}$ | $\underline{68.5_{\pm0.3}}$ | $64.1_{\pm0.9}$ | $\mathbf{69.3_{\pm0.2}}$ | $73.1_{\pm0.1}$ |
| | 0.05% | $57.6_{\pm0.7}$ | $67.0_{\pm0.2}$ | $65.9_{\pm0.2}$ | $68.4_{\pm0.4}$ | $64.8_{\pm1.1}$ | $64.7_{\pm1.4}$ | $68.9_{\pm0.3}$ | $67.9_{\pm0.3}$ | $\underline{69.8_{\pm0.3}}$ | $65.9_{\pm0.9}$ | $\mathbf{70.2_{\pm0.5}}$ | |
| | 0.1% | $59.1_{\pm0.5}$ | $68.0_{\pm0.2}$ | $66.1_{\pm0.3}$ | $68.4_{\pm0.3}$ | $67.0_{\pm0.7}$ | $66.4_{\pm0.7}$ | $69.1_{\pm0.2}$ | $70.1_{\pm0.3}$ | $\underline{71.1_{\pm0.3}}$ | $67.7_{\pm0.3}$ | $\mathbf{71.5_{\pm0.4}}$ | |
| Flickr | 0.1% | $42.0_{\pm0.7}$ | $46.5_{\pm0.4}$ | $46.9_{\pm0.1}$ | $44.5_{\pm0.4}$ | $45.3_{\pm0.7}$ | $47.0_{\pm0.1}$ | $46.8_{\pm0.0}$ | $46.6_{\pm0.2}$ | $\underline{47.1_{\pm0.1}}$ | $44.4_{\pm0.4}$ | $\mathbf{47.2_{\pm0.3}}$ | $46.8_{\pm0.2}$ |
| | 0.5% | $43.2_{\pm0.1}$ | $\underline{47.1_{\pm0.1}}$ | $\underline{47.1_{\pm0.3}}$ | $45.0_{\pm0.2}$ | $45.6_{\pm0.4}$ | $\mathbf{48.3_{\pm0.5}}$ | $47.1_{\pm0.1}$ | $47.0_{\pm0.1}$ | $47.0_{\pm0.2}$ | $45.4_{\pm0.1}$ | $47.1_{\pm0.1}$ | |
| | 1% | $44.1_{\pm0.4}$ | $47.1_{\pm0.1}$ | $47.1_{\pm0.1}$ | $44.6_{\pm0.3}$ | $43.8_{\pm1.5}$ | $\mathbf{48.4_{\pm0.9}}$ | $47.0_{\pm0.1}$ | $47.1_{\pm0.1}$ | $\underline{47.3_{\pm0.3}}$ | $45.4_{\pm0.1}$ | $47.2_{\pm0.1}$ | |
| Reddit | 0.05% | $46.6_{\pm2.3}$ | $88.0_{\pm1.8}$ | $90.5_{\pm2.1}$ | $88.9_{\pm1.2}$ | $\mathbf{91.1_{\pm1.0}}$ | $90.2_{\pm0.1}$ | $90.6_{\pm0.2}$ | $89.7_{\pm0.2}$ | $\mathbf{91.1_{\pm0.4}}$ | $90.5_{\pm0.3}$ | $90.5_{\pm0.3}$ | $94.2_{\pm0.0}$ |
| | 0.1% | $53.0_{\pm3.3}$ | $89.6_{\pm0.7}$ | $91.8_{\pm1.9}$ | $91.8_{\pm0.3}$ | $\underline{92.0_{\pm0.3}}$ | $90.6_{\pm0.9}$ | $91.4_{\pm0.1}$ | $90.0_{\pm0.3}$ | $91.4_{\pm0.2}$ | $91.3_{\pm0.2}$ | $\mathbf{93.0_{\pm0.1}}$ | |
| | 0.2% | $58.5_{\pm2.1}$ | $90.1_{\pm0.5}$ | $91.6_{\pm1.8}$ | $92.2_{\pm0.1}$ | $\underline{92.6_{\pm0.1}}$ | $91.8_{\pm0.7}$ | $91.6_{\pm0.2}$ | $90.3_{\pm0.3}$ | $91.5_{\pm0.4}$ | $91.4_{\pm0.1}$ | $\mathbf{92.9_{\pm0.2}}$ | |

Table 3: Efficiency comparison using GCN backbone (total condensation time in seconds).

| Dataset | K-Cen. | GCond | SGDD | GCDM | SimGC | EXGC | CGC | SFGC | GEOM | Ours (Pre.) | Ours |
|---|---|---|---|---|---|---|---|---|---|---|---|
| CiteSeer (r=1.8%) | 7 | 71 | 70 | 57 | 245 | 237 | 32 | 2,165 | 10,890 | 6 | 45 |
| Cora (r=2.6%) | 5 | 70 | 70 | 54 | 240 | 235 | 30 | 2,578 | 10,144 | 4 | 44 |
| PubMed (r=0.15%) | 5 | 59 | 223 | 48 | 291 | 278 | 51 | 8,060 | 26,432 | 5 | 39 |
| Arxiv (r=0.25%) | 18 | 389 | 759 | 555 | 362 | 338 | 126 | 86,553 | 104,905 | 20 | 247 |
| Products (r=0.05%) | 91 | 13,554 | 21,821 | 11,485 | 4861 | 4915 | 1093 | 1,509,397 | 1,912,105 | 104 | 2,985 |
| Flickr (r=0.5%) | 16 | 187 | 1,178 | 165 | 425 | 412 | 94 | 96,350 | 21,061 | 23 | 219 |
| Reddit (r=0.1%) | 51 | 2,665 | 12,126 | 1563 | 702 | 692 | 182 | 379,974 | 128,642 | 55 | 505 |

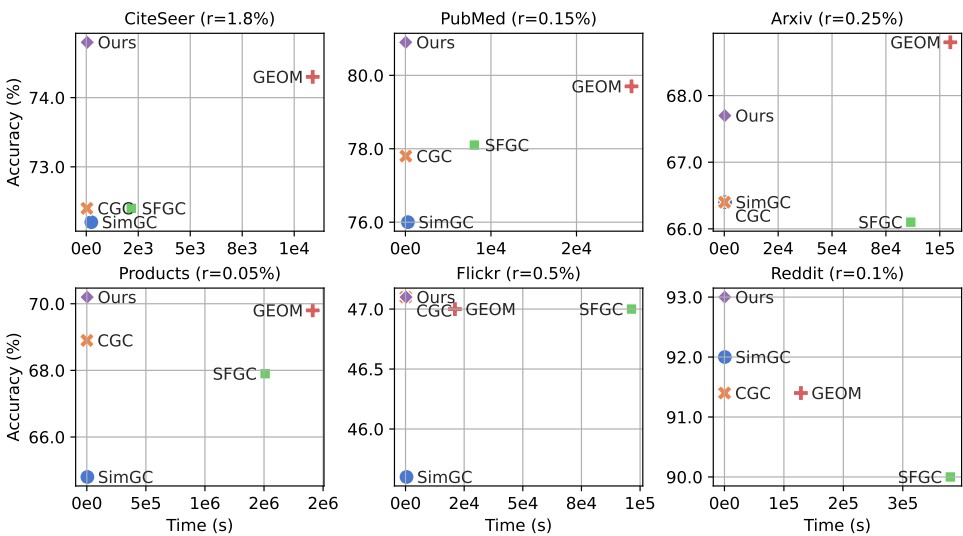

Figure 4: Evaluation accuracy versus total condensation time using GCN backbone.

Table 4: Cross-architecture transferability of condensed graphs using GCN backbone.

| Dataset | Method | MLP | SGC | GCN | GAT | ChebNet | SAGE | APPNP | Avg. | Std. |
|---|---|---|---|---|---|---|---|---|---|---|
| CiteSeer ($r = 0.90\%$) | GCond | 41.8 | 34.8 | 46.3 | 39.2 | 57.4 | 61.2 | 47.0 | 46.8 | 8.8 |
| | GCDM | 62.3 | 69.6 | 72.7 | 58.3 | 60.2 | 67.1 | 71.4 | 65.9 | 5.3 |
| | SFGC | 64.4 | 64.9 | 70.4 | 70.0 | 69.1 | 69.5 | 70.8 | 68.4 | 2.5 |
| | Ours | 66.5 | 70.9 | 73.4 | 73.4 | 72.8 | 72.6 | 72.1 | **71.7** | 2.3 |
| Cora ($r = 1.30\%$) | GCond | 67.7 | 72.6 | 79.5 | 80.7 | 60.0 | 78.6 | 79.0 | 74.0 | 7.2 |
| | GCDM | 65.3 | 78.5 | 80.2 | 80.1 | 58.4 | 77.5 | 79.3 | 74.2 | 8.1 |
| | SFGC | 68.2 | 76.2 | 80.4 | 79.8 | 62.1 | 77.6 | 81.6 | 75.1 | 6.7 |
| | Ours | 70.5 | 79.9 | 81.3 | 79.1 | 82.1 | 78.9 | 76.2 | **78.3** | 3.6 |
| PubMed ($r = 0.08\%$) | GCond | 75.1 | 55.6 | 75.0 | 77.0 | 74.3 | 77.2 | 78.0 | 73.2 | 7.3 |
| | GCDM | 73.8 | 72.9 | 75.0 | 73.7 | 70.5 | 75.3 | 76.9 | 74.0 | 1.9 |
| | SFGC | 73.6 | 76.8 | 78.5 | 76.6 | 77.2 | 76.7 | 78.9 | **76.9** | 1.6 |
| | Ours | 74.2 | 76.6 | 76.1 | 76.3 | 77.3 | 77.5 | 76.7 | 76.4 | 1.0 |
| Arxiv ($r = 0.05\%$) | GCond | 39.2 | 58.0 | 57.0 | 47.7 | 36.4 | 33.5 | 54.3 | 46.6 | 9.5 |
| | GCDM | 41.6 | 59.8 | 60.7 | 46.5 | 52.6 | 55.3 | 60.3 | 53.8 | 6.9 |
| | SFGC | 45.3 | 62.2 | 63.3 | 60.5 | 50.7 | 55.4 | 62.4 | 57.1 | 6.4 |
| | Ours | 46.7 | 61.6 | 65.0 | 64.4 | 63.3 | 58.4 | 53.9 | **59.0** | 6.2 |
| Products ($r = 0.025\%$) | GCond | 36.4 | 45.7 | 60.7 | 48.4 | 45.2 | 49.8 | 60.3 | 49.5 | 8.0 |
| | GCDM | 45.7 | 60.0 | 66.6 | 67.9 | 61.2 | 63.6 | 66.2 | 61.6 | 7.0 |
| | SFGC | 46.7 | 55.1 | 66.7 | 69.4 | 61.4 | 63.4 | 64.8 | 61.1 | 7.2 |
| | Ours | 46.3 | 65.9 | 65.9 | 67.6 | 67.8 | 62.2 | 62.1 | **62.5** | 7.0 |
| Flickr ($r = 0.1\%$) | GCond | 40.8 | 36.5 | 44.9 | 40.8 | 43.0 | 43.2 | 44.9 | 42.0 | 2.7 |
| | GCDM | 41.7 | 27.3 | 40.7 | 37.7 | 41.5 | 43.0 | 43.8 | 39.4 | 5.3 |
| | SFGC | 44.9 | 38.7 | 46.2 | 45.3 | 43.6 | 44.9 | 46.2 | 44.3 | 2.4 |
| | Ours | 44.1 | 45.1 | 45.3 | 45.1 | 42.4 | 43.6 | 45.4 | **44.4** | 1.0 |
| Reddit ($r = 0.05\%$) | GCond | 38.7 | 82.2 | 79.9 | 31.2 | 38.7 | 41.5 | 69.8 | 54.6 | 20.2 |
| | GCDM | 43.1 | 87.1 | 88.1 | 37.5 | 55.6 | 66.2 | 68.9 | 63.8 | 18.3 |
| | SFGC | 47.5 | 82.8 | 87.0 | 84.4 | 53.6 | 71.9 | 67.5 | 70.7 | 14.4 |
| | Ours | 39.3 | 91.1 | 90.9 | 90.5 | 61.8 | 79.1 | 66.4 | **74.1** | 18.1 |

Table 5: Ablation on precomputation components including both structure-based (Stru.) and semantic-based (Sem.) aggregation phases. The *w/o Both* variant initializes the synthetic features using randomly selected original features.

| Dataset | Ours | w/o Stru. | w/o Sem. | w/o Both |
|---|---|---|---|---|
| CiteSeer (r=1.80%) | **74.8**$_{\pm0.3}$ | 69.3$_{\pm0.6}$ | 67.8$_{\pm0.2}$ | 62.1$_{\pm0.3}$ |
| Cora (r=2.60%) | **82.9**$_{\pm1.0}$ | 74.4$_{\pm0.4}$ | 79.1$_{\pm0.7}$ | 72.2$_{\pm0.5}$ |
| PubMed (r=0.15%) | **80.9**$_{\pm0.3}$ | 77.9$_{\pm0.3}$ | 78.8$_{\pm0.4}$ | 76.1$_{\pm0.5}$ |
| Arxiv (r=0.25%) | **67.7**$_{\pm0.2}$ | 64.2$_{\pm0.6}$ | 67.3$_{\pm0.3}$ | 63.9$_{\pm0.2}$ |
| Products (r=0.05%) | **70.2**$_{\pm0.5}$ | 64.6$_{\pm0.5}$ | 65.7$_{\pm0.2}$ | 62.2$_{\pm0.7}$ |
| Flickr (r=0.5%) | **47.1**$_{\pm0.1}$ | 46.9$_{\pm0.1}$ | 47.0$_{\pm0.5}$ | 46.9$_{\pm0.6}$ |
| Reddit (r=0.10%) | **93.0**$_{\pm0.1}$ | 92.4$_{\pm0.4}$ | 92.2$_{\pm0.6}$ | 92.2$_{\pm0.2}$ |

- Our method is not only more efficient than the time-intensive trajectory-based methods but also faster than the majority of other baseline methods on most datasets. These results underscore the superior condensation efficiency of our precompute-then-adapt framework.

- A variant of our method containing only the precomputation stage (Pre.), typically taking under 60 seconds to complete, matches or surpasses 5 out of 9 baselines, as detailed in Table 2. The results illustrate the capability of the precomputation stage to achieve competitive results in a fraction of the time compared to learning-based baselines.

These observations demonstrate that our method not only achieves competitive performance but does so with markedly higher efficiency, addressing one of the key challenges in scalable graph learning applications.

### 4.4. Cross-architecture Transferability

Table 4 presents the cross-architecture transferability results of condensed graphs across different models. Our method consistently matches or outperforms the top performance across all datasets, underscoring the robustness and generalization of our framework. The ability to transfer across different architectures may be attributed to the similar filtering behavior of popular GNNs, as reported in the literature (Jin et al., 2022b; Zheng et al., 2024). In particular, our framework demonstrates outstanding transferability, which may be attributed to our direct alignment between original and synthetic features without relying on specific GNN models for performance matching.

### 4.5. Ablation on Precomputation Components

Table 5 evaluates the impact of structure-based and semantic-based phases of the precomputation stage. The results show that both the structural and semantic components contribute to the performance of the framework, particularly on transductive datasets, which reflects the importance of precom-

Table 6: Impact of diversity constraint.

| Dataset | $\gamma = 0$ | $\gamma = 0.01$ | $\gamma = 0.1$ | $\gamma = 1$ |
|---|---|---|---|---|
| CiteSeer (r=1.80%) | $72.1_{\pm 0.1}$ | $73.5_{\pm 0.4}$ | $\mathbf{74.8_{\pm 0.3}}$ | $74.2_{\pm 0.2}$ |
| Cora (r=2.60%) | $81.4_{\pm 0.2}$ | $\mathbf{82.9_{\pm 1.0}}$ | $80.3_{\pm 0.6}$ | $79.7_{\pm 0.7}$ |
| PubMed (r=0.15%) | $79.9_{\pm 0.5}$ | $80.2_{\pm 0.3}$ | $\mathbf{80.9_{\pm 0.3}}$ | $79.3_{\pm 0.4}$ |
| Arxiv (r=0.25%) | $64.6_{\pm 0.4}$ | $65.8_{\pm 0.6}$ | $\mathbf{67.7_{\pm 0.2}}$ | $66.1_{\pm 0.5}$ |
| Products (r=0.05%) | $65.9_{\pm 0.9}$ | $66.3_{\pm 0.2}$ | $\mathbf{70.2_{\pm 0.5}}$ | $68.4_{\pm 0.4}$ |
| Flickr (r=0.5%) | $45.4_{\pm 0.1}$ | $\mathbf{47.1_{\pm 0.1}}$ | $46.8_{\pm 0.5}$ | $45.9_{\pm 0.3}$ |
| Reddit (r=0.10%) | $91.3_{\pm 0.2}$ | $92.1_{\pm 0.1}$ | $\mathbf{93.0_{\pm 0.1}}$ | $92.7_{\pm 0.4}$ |

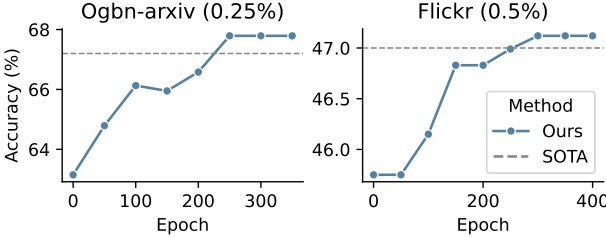

Figure 5: Effect of adaptation learning over different numbers of adaptation epochs.

putation on transductive datasets where the complete graph structure is available. We also observe that removing structural components typically results in a larger performance drop compared to the removal of semantic components. This indicates the critical role of structure-based aggregation in capturing representative features in the original graph. In conclusion, the structural and semantic components are both pivotal to achieving optimal performance in our framework, but their impact varies with the nature of the datasets.

### 4.6. Impact of Diversity Constraint

Table 6 evaluates the impact of the diversity constraint on model performance across different datasets. The hyperparameter $\gamma$ controls diversity, with higher values promoting greater diversity in the synthetic features. We observe that while the constraint generally enhances performance, excessively high values of $\gamma$ can negatively affect the results.

### 4.7. Impact of Adaptation Learning

We demonstrate the impact of the adaptation learning stage in Figure 5. On the presented large datasets, the precomputation stage (Epoch 0) produces condensed representations with sub-optimal performance. The adaptation learning further improves the representations by aligning them with the original node representations, achieving SOTA performance after sufficient training epochs.

### 4.8. Extended Analysis of Hyperparameters

As shown in Figure 6a, increasing the number of structure-based precomputation hops $K$ generally improves accuracy across datasets. A similar trend is observed in Figure 6b for the semantic aggregation size $M$. These positive correlations highlight the importance of both structure- and semantic-based aggregation modules during precomputation. Notably, performance gains begin to saturate beyond $K=2$ and $M=50$, indicating diminishing returns once the representations are sufficiently aggregated.

In contrast, Figures 6c and 6d show that varying the residual coefficient $\beta$ and the number of negative samples $S$ yields less significant changes in final accuracy. This robustness

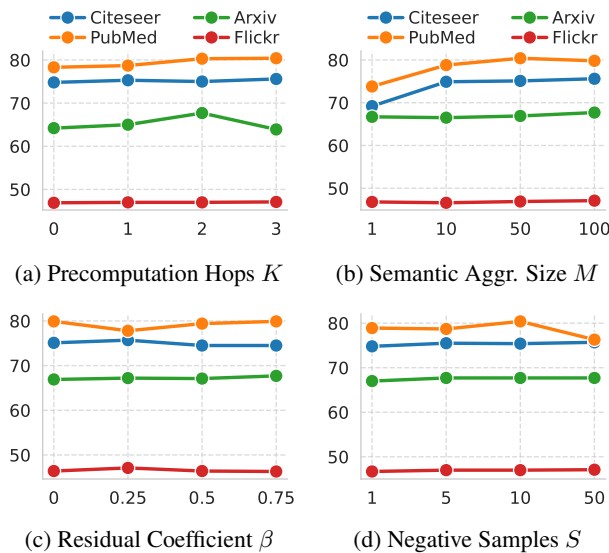

Figure 6: Impact of hyperparameters on accuracy (%).

suggests that the adaptation learning phase is relatively insensitive to these hyperparameters, reducing the need for exhaustive tuning and enhancing the method's applicability across datasets and deployment settings.

## 5. Related Work

**Graph Condensation** (GC) (Jin et al., 2022b;a; Yang et al., 2024), derived from dataset distillation (Wang et al., 2018; Zhao et al., 2021), aims to produce a smaller version of a graph while retaining information from the original. Its optimization goal is for GNNs trained on the condensed graph to perform similarly to those trained on the original. GC methods can be categorized into two classes: structured GC, which generates both node features and graph structure; and structure-free GC, which only focuses on synthesizing node features but does not generate explicit graph structure.

**Structured Graph Condensation** synthesizes graph structure using a neural network that generates links between nodes based on their features. GCond (Jin et al., 2022b) uses

a gradient matching loss between the original and condensed graphs, but its nested optimization loop limits efficiency. GCDM (Liu et al., 2022) generates smaller graphs with a distribution similar to the original graph, using a distribution matching loss measured by maximum mean discrepancy. SGDD (Yang et al., 2024) incorporates the original graph structure through optimal transport. SimGC (Xiao et al., 2024) aligns condensed and original graphs using a pretrained SGC model without external parameters for efficient graph condensation. EXGC (Fang et al., 2024) employs mean-field variational approximation and gradient-based explainability to efficiently condense large graphs. CGC (Gao et al., 2025) introduces a training-free graph condensation framework that transforms class-level distribution matching into a clustering-based class partition problem.

**Structure-Free Graph Condensation** assumes that structural information can be embedded directly into the synthetic node features, bypassing the need to generate graph structure. GCondX (Jin et al., 2022b), a variant of GCond, focuses solely on feature learning via gradient matching without the inner loop. SFGC (Zheng et al., 2024) matches training trajectories with expert guidance, and GEOM (Zhang et al., 2024) adjusts the matching range for different node difficulties. While these condensed graphs do not possess edges, they enable GNN-based node classification by introducing self-loops for each node, simplifying the training process and achieving SOTA performance.

**Graph Coarsening** methods (Cai et al., 2021; Loukas & Vandergheynst, 2018; Huang et al., 2021; Deng et al., 2019) reduce graph size by clustering original nodes into supernodes, preserving the overall structural or spectral properties while enabling more efficient downstream processing

**Coreset** methods (Sener & Savarese, 2018; Welling, 2009; Wolf, 2011) select a subset of the original nodes and retain the induced edges. K-Center (Sener & Savarese, 2018) trains a GCN on the original graph to generate embeddings, from which k-nearest nodes are sampled to form a subgraph.

Different from existing methods, our framework employs a streamlined precomputation and adaptation process that extracts and aligns features efficiently, avoiding the costly re-training in SOTA methods.

## 6. Conclusion

In this paper, we propose a framework, GCPA, for efficient graph condensation. The method consists only of a precomputation stage that performs one-time message passing and a diversity-aware adaptation stage. Compared to SOTA methods, our framework avoids costly repetitive retraining, achieving up to 2,455× faster training speed. Our framework is also effective, surpassing or matching a majority of baselines with just the precomputation stage and

achieving SOTA results with a further adaptation stage. Our framework demonstrates that precomputation is a promising solution for efficient graph condensation, and it is also flexible, as it can be further enhanced through adaptation learning. In the future, we plan to explore more precomputation techniques for graph condensation.

## Acknowledgements

This research is supported by the National Research Foundation, Singapore and Infocomm Media Development Authority under its Trust Tech Funding Initiative. Any opinions, findings and conclusions or recommendations expressed in this material are those of the author(s) and do not reflect the views of National Research Foundation, Singapore and Infocomm Media Development Authority.

## Impact Statement

This paper presents work whose goal is to advance the field of Machine Learning. There are many potential societal consequences of our work, none which we feel must be specifically highlighted here.

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

## A. Evaluation Scheme

Following GCondenser (Liu et al., 2024), we evaluate all methods using three different condensation ratios (r) for each dataset. Specifically, the condensation ratio r is defined as the fraction of condensed nodes to the total number of original nodes $N$, where $0 < r < 1$. In the transductive setting, N denotes the total node count in the entire large-scale graph, whereas in the inductive setting, N refers to the node count within the training sub-graph of the complete large-scale graph. The evaluation has two phases: (i) the condensation phase: synthesizes the condensed graph from the original graph, and (ii) the evaluation phase: the GNN is trained on the condensed graph, and the performance is evaluated on the original test nodes. We repeat the experiments five times and report the average node classification accuracy with standard deviation. The experiments are conducted on a single NVIDIA H100 GPU (80GB).

## B. Hyperparameter Settings

For our method, we tune the structure-based precomputation hops $K \in \{1, 2, 3, 4\}$, damping factor $\alpha \in \{0, 0.25, 0.5, 0.75\}$, residual coefficient $\beta \in \{0, 0.25, 0.5, 0.75\}$, diversity coefficient $\gamma \in \{0, 0.001, 0.01, 0.1, 1\}$, semantic-based aggregation size $M \in \{1, 10, 50, 100\}$, number of negative samples $S \in \{1, 5, 10, 50\}$, number of adaptation layers $\{1, 2, 3\}$, and hidden dimension of the adaptation module $\{128, 256, 512\}$. We tune all hyperparameters on the validation set. We adopt the default settings of AdamW, including learning rate $\eta = 0.001$, $\beta_1 = 0.9$, $\beta_2 = 0.999$, and $\lambda = 0.01$ for weight decay.

## C. Performance and Efficiency using SGC Backbone

Figure 7 and 8 present node classification performance and efficiency comparisons using the SGC backbone, respectively.

Table 7: Node classification performance comparison using SGC backbone (mean±std). The best and second-best results are marked in bold and underlined, respectively. *Ours (Pre.)* is the precomputation-only variant. The *Whole* column represents the performance obtained by training on the whole dataset.

| Dataset | Ratio | K-Cen. | GCond | SGDD | GCDM | SFGC | GEOM | Ours | Whole |
|---|---|---|---|---|---|---|---|---|---|
| CiteSeer | 0.9% | 52.7±0.0 | 71.9±0.6 | 71.1±0.1 | 66.0±2.2 | 65.2±0.3 | 60.1±0.2 | **72.3±0.5** | 70.3±1.0 |
| | 1.8% | 66.8±0.0 | 71.0±0.6 | 69.9±0.1 | 66.7±0.0 | 67.0±0.8 | 65.2±0.2 | **72.7±0.3** | |
| | 3.6% | 68.1±0.0 | 72.5±1.2 | 70.8±0.8 | 69.1±1.2 | 68.8±0.2 | 67.7±0.3 | **72.7±0.6** | |
| Cora | 1.3% | 63.8±0.0 | 80.6±0.1 | 62.4±5.5 | 77.0±0.4 | 73.8±1.5 | 69.2±1.2 | **80.9±0.7** | 79.2±0.6 |
| | 2.6% | 70.3±0.0 | 81.0±0.2 | 80.8±0.4 | 78.9±1.0 | 77.5±0.1 | 69.6±1.5 | **81.5±0.6** | |
| | 5.2% | 77.1±0.0 | 80.9±0.4 | 81.4±0.4 | 77.9±0.7 | 79.2±0.1 | 77.3±0.1 | **81.9±0.6** | |
| PubMed | 0.08% | 70.5±0.1 | 75.9±0.7 | 76.4±0.9 | 73.3±1.2 | 73.9±0.5 | 73.8±0.3 | **76.6±0.5** | 76.9±0.1 |
| | 0.15% | 75.8±0.0 | 75.2±0.0 | **78.0±0.3** | 74.7±0.6 | 75.8±0.2 | 77.4±0.4 | 76.9±0.6 | |
| | 0.3% | 75.7±0.0 | 75.7±0.0 | 76.1±0.1 | 76.5±1.1 | 75.8±0.0 | 75.8±0.4 | **76.8±0.4** | |
| Arxiv | 0.05% | 51.8±0.2 | 65.5±0.0 | 64.5±0.9 | 60.8±0.1 | 66.1±0.2 | 62.0±0.5 | **67.2±0.4** | 68.8±0.0 |
| | 0.25% | 58.2±0.0 | 66.5±0.5 | 66.4±0.3 | 62.7±0.9 | 66.7±0.3 | 62.8±0.7 | **67.3±0.2** | |
| | 0.5% | 60.3±0.0 | 67.2±0.1 | 66.9±0.3 | 62.4±0.2 | 66.4±0.3 | 63.6±0.3 | **67.3±0.1** | |
| Products | 0.025% | 48.6±0.6 | 64.0±0.2 | 64.9±0.1 | 57.7±0.2 | 62.2±0.1 | 61.1±0.4 | **65.0±0.5** | 64.7±0.1 |
| | 0.05% | 52.2±0.7 | 64.0±0.1 | 62.3±0.2 | 58.2±0.3 | 62.2±0.2 | 62.4±0.2 | **65.1±0.4** | |
| | 0.1% | 55.4±0.4 | 64.4±0.4 | 64.3±0.3 | 60.8±0.2 | 61.9±0.2 | 63.1±0.2 | **65.0±0.4** | |
| Flickr | 0.1% | 34.5±0.1 | 43.7±0.5 | 43.6±0.3 | 40.3±0.0 | 45.3±0.7 | 33.6±0.4 | **45.6±0.3** | 44.2±0.0 |
| | 0.5% | 36.1±0.0 | 42.2±0.2 | 41.6±1.6 | 40.8±0.1 | 45.7±0.4 | 37.4±0.2 | **46.5±0.2** | |
| | 1% | 36.5±0.0 | 41.1±0.8 | 43.2±0.4 | 42.7±0.4 | 46.1±0.5 | 38.1±0.2 | **46.8±0.2** | |
| Reddit | 0.05% | 54.0±0.1 | 89.7±0.6 | 90.5±0.3 | 90.3±0.8 | 90.9±0.2 | 59.4±1.5 | **91.5±0.7** | 93.2±0.0 |
| | 0.1% | 78.6±0.0 | 91.8±0.2 | 91.9±0.0 | 88.1±2.8 | 92.6±0.2 | 81.7±0.7 | **92.6±0.1** | |
| | 0.2% | 83.8±0.0 | 92.1±0.3 | 86.3±5.6 | 91.7±0.2 | 92.6±0.3 | 86.7±0.1 | **92.7±0.2** | |

## D. Visualization

Figure 7 displays visualization results between SFGC condensed features and ours. Our condensed graphs exhibit clear clustering patterns on all presented datasets with minimal inter-class mixing, in contrast to the SFGC graphs, which show less distinct class separation. The comparison is more evident on larger datasets such as Ogbn-arxiv and Flickr, where SFGC fails to produce clear clustering patterns. To quantify these clustering patterns, we follow previous work (Zhang et al., 2024)

Table 8: Efficiency comparison using SGC backbone (total condensation time in seconds).

| Dataset | K-Center | GCond | SGDD | GCDM | SFGC | GEOM | Ours (Pre.) | Ours |
|---|---|---|---|---|---|---|---|---|
| CiteSeer (r=1.8%) | 5 | 42 | 51 | 47 | 1,652 | 6,920 | 6 | 26 |
| Cora (r=2.6%) | 4 | 42 | 48 | 47 | 2,011 | 6,031 | 4 | 21 |
| PubMed (r=0.15%) | 4 | 34 | 204 | 42 | 7,555 | 22,201 | 5 | 40 |
| Arxiv (r=0.25%) | 6 | 283 | 1,485 | 242 | 78,586 | 84,356 | 20 | 71 |
| Products (r=0.05%) | 44 | 2,011 | 2,007 | 1,545 | 1,357,845 | 1,687,718 | 104 | 586 |
| Flickr (r=0.5%) | 5 | 177 | 300 | 258 | 99,254 | 19,202 | 23 | 56 |
| Reddit (r=0.1%) | 7 | 508 | 9,203 | 505 | 360,327 | 100,354 | 55 | 91 |

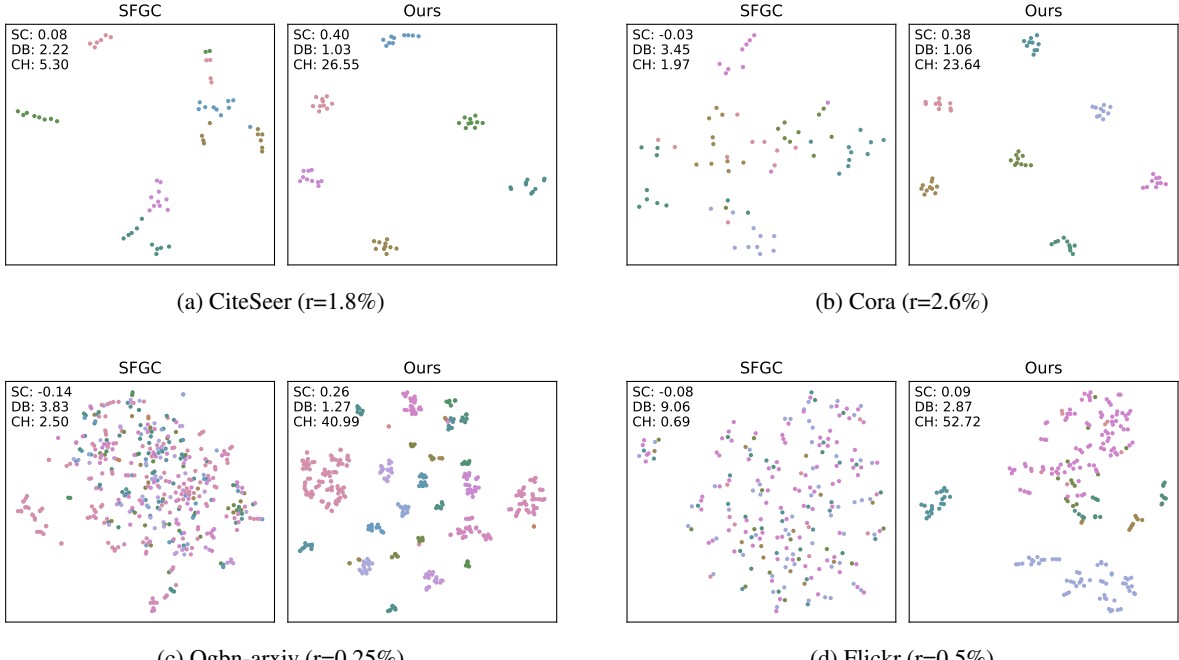

(a) CiteSeer (r=1.8%)  (b) Cora (r=2.6%)

(c) Ogbn-arxiv (r=0.25%)  (d) Flickr (r=0.5%)

Figure 7: The t-SNE visualization of synthetic node features using GCN backbone. The node classes are represented by colors. The clustering metrics including Silhouette Coefficient (SC↑), Davies-Bouldin Index (DB↓), and Calinski-Harabasz Index (CH↑) are reported for each plot. The arrows ↑ and ↓ denote that a higher value indicates better clustering pattern for SC and CH, while a lower value indicates better clustering for DB.

to utilize clustering metrics including the Silhouette Coefficient (Rousseeuw, 1987), the Davies-Bouldin Index (Davies & Bouldin, 1979), and the Calinski-Harabasz Index (Caliński & Harabasz, 1974), all of which indicate that our condensed graphs demonstrate better clustering patterns. The visualization results show that our framework effectively optimizes the condensed features, forming robust representations to preserve the original graph's classification capabilities.

## E. Structure-free Features via Precomputation

During the precomputation stage, we transform the raw features to structure-free features via precomputation. We show that when using SGC as the backbone GNN, the precomputed features coupled with an identity adjacency matrix are equivalent to the raw features coupled with the original graph structure. We start by defining the SGC network on the original graph:

$$\text{SGC}(\mathbf{X}, \mathbf{A}; \mathbf{\Theta}) = \left( \tilde{\mathbf{D}}^{-\frac{1}{2}} \tilde{\mathbf{A}} \tilde{\mathbf{D}}^{-\frac{1}{2}} \right)^{K} \mathbf{X} \mathbf{\Theta}, \tag{6}$$

where $\mathbf{X}$ is the raw node features, $\tilde{\mathbf{A}} = \mathbf{A} + \mathbf{I}$ represents the adjacency matrix with self-loops, $\tilde{\mathbf{D}}$ denotes the degree matrix of $\tilde{\mathbf{A}}$, $K$ is the number of propagation layers, and $\mathbf{\Theta}$ is the weight matrix.

Then, we revisit the feature precomputation introduced in Equation 2 when $\alpha = 0$:

$$\mathbf{X}' = \left(\tilde{\mathbf{D}}^{-\frac{1}{2}}\tilde{\mathbf{A}}\tilde{\mathbf{D}}^{-\frac{1}{2}}\right)^{K}\mathbf{X}, \tag{7}$$

where $\mathbf{X}'$ denotes the precomputed features, which is the result of applying the same transformation as in the SGC but isolated from the learning weights $\boldsymbol{\Theta}$.

As a result, SGC with precomputed features and identity adjacency matrix becomes:

$$\mathrm{SGC}(\mathbf{X}', \mathbf{I}; \boldsymbol{\Theta}) = \left(\tilde{\mathbf{D}}^{-\frac{1}{2}}\tilde{\mathbf{I}}\tilde{\mathbf{D}}^{-\frac{1}{2}}\right)^{K}\mathbf{X}'\boldsymbol{\Theta} = \mathbf{X}'\boldsymbol{\Theta} = \left(\tilde{\mathbf{D}}^{-\frac{1}{2}}\tilde{\mathbf{A}}\tilde{\mathbf{D}}^{-\frac{1}{2}}\right)^{K}\mathbf{X}\boldsymbol{\Theta}, \tag{8}$$

Therefore, we draw the equivalence between SGC computation on the original graph and the structure-free precomputed features:

$$\mathrm{SGC}(\mathbf{X}', \mathbf{I}; \boldsymbol{\Theta}) = \mathrm{SGC}(\mathbf{X}, \mathbf{A}; \boldsymbol{\Theta}) \tag{9}$$

The equivalence shows that although the original features and condensed features are differently distributed, they perform equivalently when coupled with their corresponding graph structure using the SGC backbone. Drawing inspiration from this equivalence under the SGC backbone, our framework focuses on initializing and refining features in the precomputed feature space, enabling effective training of message-passing GNNs on the structure-free condensed graphs.

