# OpenReview forum: "Adapting Precomputed Features for Efficient Graph Condensation"
_ICML.cc/2025/Conference — ICML 2025 poster_

### Official Review · Reviewer_rs1P · 2025-03-10

**Overall Recommendation:** 2

**Summary:**

To address the efficiency issue in graph condensation (GC), this paper proposes GCPA, a two-stage framework comprising precomputation and diversity-aware adaptation. The precomputation stage aggregates structural and semantic information for competitive performance, while the adaptation stage refines features via class-wise alignment with minimal cost. Experiments on seven benchmarks confirm its superior efficiency.

**Claims And Evidence:**

Yes

**Essential References Not Discussed:**

The key contribution of the paper is efficient GC. However, some recent proposed method are not discussed, including SimGC[1], EXGC[2], and CGC[3].

[1] Xiao, Z., Wang, Y., Liu, S., Wang, H., Song, M., & Zheng, T. Simple graph condensation. In ECML-PKDD 2024

[2] Fang, J., Li, X., Sui, Y., Gao, Y., Zhang, G., Wang, K., ... & He, X. (2024, May). Exgc: Bridging efficiency and explainability in graph condensation. In WWW2024

[3] Gao, X., Ye, G., Chen, T., Zhang, W., Yu, J., & Yin, H. (2024). Rethinking and accelerating graph condensation: A training-free approach with class partition. arXiv preprint arXiv:2405.13707.

**Experimental Designs Or Analyses:**

I have reviewed all aspects of the experiment section, including the experimental setup, performance evaluation, efficiency evaluation, transferability evaluation, ablation study, and parameter analysis. Overall, the experimental design and analyses are reasonable. However, some important baselines are missing.

**Methods And Evaluation Criteria:**

Yes

**Other Comments Or Suggestions:**

N/A

**Other Strengths And Weaknesses:**

Strength:
1. The paper is generally well-written.
2. The efficiency improvement in GC appears to be significant.

Weakness:
1. The novelty of the paper is poor.
2. A lack of theoratical analysis on why the method is effective.
3. Some important baselines are missing.

**Questions For Authors:**

1. The novelty of the paper is limited. It seems that the structure-precomputation only uses the graph diffusion operators which is widely used to perform message passing, and semantic precomputation is a simple average operation. Could the author further clarify the novelty of this paper.

2. The proposed method appears to be very simple. Could the author provide a theoratical analysis on how the proposed GCPA can achieve comparative or even superior data utility (i.e., classification performance) than the previous GC methods?

3. Why the author do not compare GCPA with recently proposed efficient GC methods, including SimGC[1], EXGC[2], and CGC[3].

[1] Xiao, Z., Wang, Y., Liu, S., Wang, H., Song, M., & Zheng, T. Simple graph condensation. In ECML-PKDD 2024

[2] Fang, J., Li, X., Sui, Y., Gao, Y., Zhang, G., Wang, K., ... & He, X. (2024, May). Exgc: Bridging efficiency and explainability in graph condensation. In WWW2024

[3] Gao, X., Ye, G., Chen, T., Zhang, W., Yu, J., & Yin, H. (2024). Rethinking and accelerating graph condensation: A training-free approach with class partition. arXiv preprint arXiv:2405.13707.

**Relation To Broader Scientific Literature:**

The contributions of this paper are related to previous studies on graph neural networks (GNN) [1], and data condensation (DC)[2].

[1] Wu, Z., Pan, S., Chen, F., Long, G., Zhang, C., & Philip, S. Y.  A comprehensive survey on graph neural networks. TNNLS 2020.

[2] Cui, J., Wang, R., Si, S., & Hsieh, C. J. (2022). Dc-bench: Dataset condensation NeurIPS2022

**Theoretical Claims:**

There is no theoretical claims and proofs in this paper.

---

> ### Author Rebuttal · Authors · 2025-04-01
>
> We thank the reviewer for the examination of our work and the thoughtful comments provided. Kindly find our responses to the raised comments and questions below.
>
> **Q1: Some recently proposed efficient GC are not discussed, including SimGC[1], EXGC[2], and CGC[3].**
>
> We thank the reviewer for highlighting the recent efficient GC methods. We set the **smallest condensation ratios** and present **accuracy** with **total running time**. Considering the inconsistent time measurements (e.g., CGC reports only the condensation time), we uniformly run evaluation to measure total running time. Our method consistently outperforms the baselines, while the efficient baselines underperform GEOM. We will revise the manuscript to incorporate these methods. (For fair comparison, we adopt the new baselines’ finer hyperparameter search and update GCPA results accordingly, which slightly differ from those in the paper.)
>
> |Dataset|SimGC|EXGC|CGC|GEOM|GCPA|
> |---|---|---|---|---|---|
> |Citeseer|73.8 (245s)|69.2 (237s)|72.5 (32s)|73.0 (6,920s)|**75.4** (45s)|
> |Cora|80.8 (240s)|82.0 (235s)|82.7 (30s)|82.5 (6,031s)|**82.9** (44s)|
> |Arxiv|63.6 (362s)|57.6 (338s)|64.1 (126s)|65.5 (84,356s)|**67.2** (247s)|
> |Products|63.3 (4,861s)|62.1 (4,915s)|68.0 (1,093s)|68.5 (1,687,718s)|**69.3** (2,985s)|
> |Flickr|45.3 (425s)|47.0 (412s)|46.8 (94s)|47.1 (19,202s)|**47.2** (219s)|
> |Reddit|91.1 (702s)|90.2 (692s)|90.6 (182s)|91.1 (100,354s)|**91.3** (505s)|
> |AvgDiff|**-2.6**|**-4.2**|**-1.4**|**-0.9**|-|s
>
> **Q2: The novelty of the paper is limited. It seems that the structure-precomputation only uses the graph diffusion operators which is widely used to perform message passing, and semantic precomputation is a simple average operation.**
>
> Our work is motivated by the need to maintain strong structural and semantic guidance from the precomputed features. In many existing methods, precomputed features serve only as **temporary constraints**—initializing the learnable condensed features $Z'$—which can then vanish from the original signal as training progresses. In contrast, our approach employs a **permanent constraint**, where the precomputed features $\hat{X}'$ continuously guide the adaptation: $Z' = f_{adapt}(\hat{X}')$, ensuring their influence remains intact throughout training.
>
> We illustrate this distinction using a variant of GCPA, replacing $f_{adapt}$ with learnable condensed features $Z'$. The performance drop indicate that our permanent constraint is effective in preserving critical information.
>
> ||Arxiv|Flickr|
> |---|---|---|
> |GCPA-Variant (Temporary constraint with learnable $Z'$)|66.9|46.2|
> |GCPA (Permanent constraint with learnable $f_{adapt}$ and fixed $\hat{X}'$)|**67.7**|**47.1**|
>
> **Q3: A lack of theoratical analysis on why the method is effective. The proposed method appears to be very simple. Could the author provide a theoratical analysis on how the proposed GCPA can achieve comparative or even superior data utility (i.e., classification performance) than the previous GC methods?**
>
> We appreciate the reviewer’s question and provide theoretical insight below.
>
> **[Under SGC model, graph condensation reduces to feature set condensation]**
>
> In Appendix G, we establish that, for the Simple Graph Convolution (SGC) model, the node embeddings can be precomputed as $X'=(\tilde{D}^{-0.5}\,\tilde{A}\,\tilde{D}^{-0.5})^KX$. Hence, replacing $A$ by an identity adjacency $I$ while using $X'$ is equivalent to using $A$ with the original features $X$. Thus, condensing a graph under SGC is essentially condensing the precomputed features $X'$.
>
> **[Contrastive loss increases mutual information]**
>
> Our contrastive loss uses logistic cross-entropy to distinguish positive (same-class) pairs from negative (randomly sampled) pairs. This is known to maximize a lower bound on $\mathrm{JS}(p^+\|p^-)$, where $p^+$ and $p^-$ are the positive and negative pair distributions. As $\mathrm{JS}(p^+\|p^-)$ increases, $X'$ and the condensed features $Z'$ become more class-discriminative, thereby increasing their mutual information on class-relevant signals. Combined with the SGC equivalence, it follows that condensing $X'$ under this contrastive loss effectively preserves and amplifies the class-relevant information in the original graph features $X$.
>
> To support the claim on the effectiveness of our method, we present KSG mutual information (MI) [a] with accuracies below. The results show a clear increase in mutual information during both stages, along with improved accuracies. This trend highlights the importance of class separation in the adaptation process, thereby contributing to the strong performance of GCPA by helping to preserve class boundaries.
>
> |Stage|Arxiv-MI|Arxiv-Acc|Flickr-MI|Flickr-Acc|
> |---|---|---|---|---|
> |Precomputation|0.016|64.6|0.011|45.4|
> |Adaptation at epoch 10|0.044|65.3|0.016|45.9|
> |Adaptation at epoch 50|0.397|66.8|0.148|46.3|
> |Adaptation at epoch 100|0.567|67.2|0.347|46.8|
>
> [a] Estimating mutual information

---

### Official Review · Reviewer_nZN9 · 2025-03-11

**Overall Recommendation:** 3

**Summary:**

This paper propose the GCPA method, which not only bring the unbelievable efficiency into the graph condensation process but also gains considerable results, for example, for the Ogbn-products dataset, the conventional trajectory method calls for 452 hours in collecting the trajectories, but the GCPA only cost much less time, while even achieving the SOTA results.

**Claims And Evidence:**

As far as I can see, this paper mainly claims that they achieve better performance with 96x to 2,455x faster than the SOTA methods.
For the evidence, they provide the detailed experiments in Tab. 2, and provide the time comparison in Fig.4.

**Essential References Not Discussed:**

No.

**Ethical Review Concerns:**

No.

**Experimental Designs Or Analyses:**

Yes.

**Methods And Evaluation Criteria:**

This is what I concern most, because if we only see the individual part of the precomputation stage, each of them seems like a normal step that was widely used in the previous literature. For example, in the CTRL[A], the authors discuss the different ways of initial sets selection. Therefore, from my understanding, the authors just change the expensive matching process to a simple contrastive learning process.
	Then, my questions are listed as follow:
If you can ablate the adaptation stage to the other expensive matching processes? Because the precomputation stage seems like a normal trick to me? (I see the experiments on the appendix D, maybe it is associated with my question? But I cannot fully understand them.)
Could you give a deep explanation of the effectiveness of your methods? The current one is more like a technique report, I didn’t see the clear motivation why you use such a precomputation stage and contrastive learning technique?
I am not fully convinced by the story, but I appreciate the simple idea and impressive experiment results. I’ll give the borderline accept for now, and decide later depending on the answer in the rebuttal period.

[A] CTRL: GRAPH CONDENSATION VIA CRAFTING RATIONAL TRAJECTORY MATCHING. Zhang et.al. Arxiv 2023.

**Other Comments Or Suggestions:**

No.

**Other Strengths And Weaknesses:**

Strengths:
1. Good results, comprehensive comparsion on addtional PubMed, Products datasets.

Weakness:
1. Not that convincing, just a combination of the data process to me, better inspiration is expected.

**Questions For Authors:**

See above.

**Relation To Broader Scientific Literature:**

It is a big innovation since the authors significantly reduce the condensing time but hold considerable results.

**Theoretical Claims:**

I didn’t see any proof.

---

> ### Author Rebuttal · Authors · 2025-04-01
>
> We thank the reviewer for the examination of our work and the thoughtful comments provided. Kindly find our responses to the raised comments and questions below.
>
> **Q1: Can you ablate the adaptation stage to the other expensive matching processes? The precomputation stage seems like a normal trick.**
>
> Thank you for the suggestion. We would like to clarify that our framework—precomputation followed by adaptation—is fundamentally different from existing methods, which typically follow an initialize-then-learn paradigm. The key difference lies in where **message passing**—the operation that captures graph structural information—is performed, as shown below:
>
> Existing methods (e.g., CTRL, GEOM):
>
> - Initialization: random sampling, clustering, etc.
> - Learning: GNN training with repetitive **message passing** (the main source of computational cost)
>
> Ours:
>
> - Precomputation: one-time **message passing**
> - Adaptation: MLP training (without message passing)
>
> Our framework replaces the costly, repeated message passing (in the learning stage) in existing methods with a one-time message passing step (in the precomputation stage), achieving both efficiency and strong performance. This is not just a simple preprocessing trick, and we believe it constitutes a meaningful contribution.
>
> As for replacing adaptation with existing matching processes, such changes may lead to redundant message passing (since precomputation already involves message passing, and existing matching processes involve repeated message passing during learning) and defeat the purpose of our efficiency gains. Nonetheless, we believe it deserves future exploration.
>
> **Q2: I cannot fully understand the setting of Appendix D.**
>
> Thank you for the question. In Appendix D, we highlight the effect of precomputation by comparing two settings:
>
> - GCPA (Ours):
>     1. Structurally precompute features $H$
>     2. Semantically precompute condensed features $\hat{X}'$
>     3. Adapt condensed features $Z'=f_{adapt}(\hat{X}')$ with precomputed features $H$
> - GCPA without Precomputation (Ours with Random Initialization):
>     1. Randomly initialize condensed features $X_{rand}'$,
>     2. Adapt condensed features $Z'=f_{adapt}(X_{rand}')$ with original features $X$
>
> Table 9 shows that removing precomputation leads to loss of structural information, and hence significantly worse performance. We conclude that precomputation is not merely an initialization step—it plays a critical role by embedding structural information and guiding adaptation learning. This ablation confirms that precomputation meaningfully contributes to performance, beyond what adaptation alone can achieve.
>
> **Q3: A deep explanation of the effectiveness of your methods? Clear motivation why you use such a precomputation stage and contrastive learning technique? Not that convincing, just a combination of the data process to me, better inspiration is expected.**
>
> The two stages–precomputation and adaptation–are motivated by the need to maintain strong structural and semantic guidance from the precomputed features. In many existing methods, precomputed features serve only as **temporary constraints**—initializing the learnable condensed features $Z'$—which can then vanish from the original signal as training progresses. In contrast, our approach employs a **permanent constraint**, where the precomputed features $\hat{X}'$ continuously guide the adaptation: $Z' = f_{adapt}(\hat{X}')$, ensuring their influence remains intact throughout training.
>
> We illustrate this distinction using a variant of GCPA, replacing $f_{adapt}$ with learnable condensed features $Z'$. The performance drop indicate that our permanent constraint is effective in preserving critical information.
>
> ||Arxiv|Flickr|
> |---|---|---|
> |GCPA-Variant (Temporary constraint with learnable $Z'$)|66.9|46.2|
> |GCPA (Permanent constraint with learnable $f_{adapt}$ and fixed $\hat{X}'$)|**67.7**|**47.1**|
>
> Besides, we provide theoretical insights on the effectiveness of our method from the perspective of mutual information. We kindly refer to **Reviewer rs1P Q3** for this discussion.

---

### Official Review · Reviewer_sWb8 · 2025-03-13

**Overall Recommendation:** 3

**Summary:**

This paper introduces Graph Condensation via a Precompute-then-Adapt Approach (GCPA), an efficient method for condensing large-scale graphs to accelerate Graph Neural Network (GNN) training. The proposed framework is more computationally efficient than trajectory matching methods and instead consists of two stages: (1) a precomputation stage, which extracts structural and semantic information using a single pass of message passing, and (2) an adaptation stage, which refines the synthetic features using class-wise feature alignment and diversity maximization. The method is evaluated across multiple benchmark datasets, showing up to 2,455× speedup while achieving performance comparable to or better than state-of-the-art (SOTA) graph condensation methods.

## update after rebuttal
As I noted during the rebuttal, I am still concerned about the hyperparameters, and the core contribution driving performance seems relatively straightforward. I will therefore maintain my score.

**Claims And Evidence:**

The authors make strong claims regarding the efficiency of their approach. This claim is backed up well in the paper, e.g., in Figures 2, 4.

**Essential References Not Discussed:**

No, I didn't identify any essential references that were missing.

**Experimental Designs Or Analyses:**

The authors’ experimental design follows standard practice and aligns with previous research.

**Methods And Evaluation Criteria:**

The authors’ evaluation protocol follows standard practice and aligns with previous research.

**Other Comments Or Suggestions:**

See questions below.

**Other Strengths And Weaknesses:**

**Strengths**:

1. The proposed technique noticeably reduces the time required for the graph condensation process and effectively addresses the issue of repeated training consumption in large-scale graphs for existing structure-free methods, which seems reasonable to me.

2. The proposed technique performs well on benchmark datasets with various condensation ratios, even during initial precomputation.

**Weaknesses**:
1. The method introduces a large number of hyperparameters beyond the standard ones used for training (e.g., hidden dimensions, number of layers, etc.), including:
* $K$ – the number of precomputation hops
* $\alpha$ – the damping factor
* $\beta$ – the residual coefficient
* $\gamma$ – the diversity coefficient
* $M$ – the semantic-based aggregation size
* $S$ – the number of negative samples

However, apart from $\gamma$, the authors do not discuss or demonstrate the effect of these hyperparameters on the method’s performance. This is particularly concerning given the large number of hyperparameters involved.

2. The precomputation stage is mostly without learning, with the exception of Equation 4, where $f_{\text{adapt}}$ operates only on the (updated) node features. I understand how this could be advantageous, but it also introduces certain limitations. Does it truly make sense to coarsen the graph using only non-learnable message passing, with the learning component restricted to (the updated) node features? I appreciate the authors' acknowledgment of this issue and their attempt to address it in the discussion (lines 172–186) via Equation 4. However, this approach still seems somewhat problematic to me.

3. The results, as shown in Table 2 for example, generally surpass the baselines. However, the margin between the best result of the proposed method and the strongest baseline remains relatively small in almost all cases, which makes the method less compelling.

**Questions For Authors:**

1. If I understand correctly, the condensed graph has no edges. So why would it make sense to apply a GNN to train on the condensed graph?

2. In Table 3, does the GCN backbone play a role in the condensation process of the proposed method, or is it only relevant to the baselines? If I understand correctly, the condensation is determined by Equations 2, 3, and 4, none of which involve a GNN.

3. In Table 2, I notice that in some cases, the results after the precomputation stage are very similar to those after both stages (precomputation and adaptation). Could you provide some insight into why the adaptation stage appears to have only a minor impact in these cases? Intuitively, I would have expected adaptation to play a more significant role than precomputation.

**Relation To Broader Scientific Literature:**

The authors include a "Related Work" section that effectively discusses relevant topics.

**Theoretical Claims:**

The authors don’t provide any theoretical claims.

---

> ### Author Rebuttal · Authors · 2025-04-01
>
> We thank the reviewer for the examination of our work and the thoughtful comments provided. Kindly find our responses to the raised comments and questions below.
>
> **Q1: Code cannot be opened.**
>
> We apologize for the inconvenience. It appears there was a temporary issue. We have refreshed the repository and it is now accessible.
>
> **Q2: The authors do not discuss the effect of many hyperparameters.**
>
> Thank you for pointing this out. In the extended experiments, we tune the key hyperparameters below on the validation set and analyze their impact:
>
> - Semantic aggregation size $M$
> - Negative sample size $S$
> - Damping factor $\alpha$
> - Precomputation hops $K$
> - Residual coefficient $\beta$
>
> The results below show that the model is generally robust to these settings. Some hyperparameters ($M$, $S$) benefit from larger values, while others ($\alpha$) have stable optimal choices. A few others ($K$, $\beta$) require more careful tuning. We will clarify these findings in the revised version.
>
> |$M$|Arxiv|Flickr|
> |---|---|---|
> |1|66.7|46.8|
> |10|66.5|46.6|
> |50|66.9|46.9|
> |100|67.7|47.1|
>
> |$S$|Arxiv|Flickr|
> |---|---|---|
> |1|67.0|46.7|
> |5|67.7|47.0|
> |10|67.7|47.0|
> |50|67.7|47.1|
>
> |$\alpha$|Arxiv|Flickr|
> |---|---|---|
> |0|67.0|45.3|
> |0.25|67.7|47.1|
> |0.5|66.5|47.0|
> |0.75|65.6|47.0|
>
> |$K$|Arxiv|Flickr|
> |---|---|---|
> |0|64.2|46.9|
> |1|65.0|47.0|
> |2|67.7|47.0|
> |3|63.9|47.1|
> |4|63.8|47.0|
>
> |$\beta$|Arxiv|Flickr|
> |---|---|---|
> |0|66.9|46.4|
> |0.25|67.2|47.1|
> |0.5|67.1|46.4|
> |0.75|67.7|46.3|
>
> **Q3: Does it truly make sense to coarsen the graph using only non-learnable message passing, with the learning component restricted to (the updated) node features?**
>
> Thank you for the thoughtful comment. We agree that relying on non-learnable message passing can introduce limitations and may not generalize to all datasets. However, in many real-world datasets—such as those used in graph condensation benchmarks—non-learnable message passing combined with MLPs (e.g., SIGN [a]) have shown competitive or even SOTA performance. A likely reason is that these benchmark datasets exhibit strong homophily or stable neighborhood patterns, allowing fixed message passing to capture key structural information effectively.
>
> Our method follows a similar principle—non-learnable message passing combined with MLP adaptation—and achieves strong results, suggesting that this design can be effective in practice.
>
> **Q4: The margin between the best result of the proposed method and the strongest baseline remains relatively small in almost all cases.**
>
> Thank you for the observation. Our method is primarily designed for efficiency rather than solely maximizing performance. Given that current SOTA methods can be impractical (e.g., requiring up to 452 hours), our goal is to offer a more efficient alternative while aiming to match, not necessarily surpass, SOTA performance. Notably, our approach achieves significant speedup and even delivers leading results, which we find both promising and encouraging.
>
> **Q5: The condensed graph has no edges, so why would it make sense to apply GNN on it?**
>
> Your are correct that the condensed graph has no edges. This setup was first investigated in SFGC [b], where only self-loops are available for GNN message passing. Although counterintuitive, these methods—including ours—have shown superior performance. A potential reason is that the model learns to encode structural information into the condensed features for GNN to learn effectively, even without edges. We follow this established setting and believe it’s an interesting direction for further investigation.
>
> **Q6: Does GCN backbone play a role in the condensation process of the proposed method?**
>
> You are correct that GCPA does not rely on GCN during condensation. We appreciate your observation and will clarify this in revised version:
>
> - Condensation: All methods except GCPA use GCN to guide learning.
> - Evaluation: All methods including GCPA use GCN for evaluation.
>
> **Q7: Why adaptation has only a minor impact in some cases? Intuitively, I would have expected adaptation to play a more significant role than precomputation.**
>
> Thank you for the observation. The impact of adaptation indeed varies in a large range, affected by the quality of the precomputed features, which in turn is influenced by factors like data noises. We illustrate this by adding noise below. When the data is clean, precomputed features are already strong, so adaptation yields modest improvements. When we add noise and degrade the precomputed features, the adaptation becomes much more beneficial.
>
> |$\sigma$ (Gaussian noise)|PubMed (Precomp.)|PubMed (Adapt)|Improve|
> |---|---|---|---|
> |0|79.7|80.5|+0.8|
> |0.01|75.7|77.4|+1.7|
> |0.05|57.4|61.4|+4.0|
> |0.1|49.8|57.1|+7.3|
> |1|42.7|54.4|+11.7|
>
> [a] SIGN: Scalable Inception Graph Neural Networks
>
> [b] Structure-free Graph Condensation: From Large-scale Graphs to Condensed Graph-free Data

---

> > ### Comment · Reviewer_sWb8 · 2025-04-03
> >
> > I appreciate the authors’ responses and the additional experiments they conducted. However, my concerns regarding the hyperparameters still remain, these experiments over the hyperparameters were only carried out on two datasets, which offers limited validation and is not entirely convincing. Having said that, I understand the constraints of the rebuttal phase and the difficulty of running large-scale experiments in such a short time.
> >
> > Regarding my concern about the limited impact of the learned adaptation, I find the response not convincing enoug. Demonstrating that adaptation helps when noise is added is kind of expected, given that the adaptation is being learned. This suggests that the precomputation step—while seems pretty trivial—is doing most of the heavy lifting.
> >
> > To summarize, the precomputation appears to be the most effective part of the proposed method. While the approach may be more efficient, it heavily depends on hyperparameters, which could pose practical challenges in real-world applications.
> >
> > Therefore, I maintain my score of a weak accept. The paper presents some valuable ideas, but the core contribution that drives performance seems relatively straightforward in my opinion.

---

> > > ### Author Response · Authors · 2025-04-04
> > >
> > > We sincerely appreciate your thoughtful comments and your engagement with us.
> > >
> > > **[Addressing Hyperparameter Concerns]**
> > >
> > > Thank you for your thoughtful feedback and for acknowledging the additional experiments we provided. We understand and respect your continued concerns regarding the scope of our hyperparameter analysis. Given the constraints of the rebuttal phase, we aimed to provide a representative but manageable set of experiments across two datasets to illustrate the consistency of our method’s behavior. We fully agree that a broader evaluation would strengthen the validation, and we will expand this in the updated version of the paper.
> > >
> > > **[Clarifying the Role of the Adaptation Stage]**
> > >
> > > Regarding the concern that the adaptation stage might contribute less than the precomputation step, we would like to clarify that while precomputation is indeed a simple and efficient component, it is the adaptation stage that consistently provides meaningful performance gains across a variety of datasets and settings, achieving SOTA performance. We present the quantitative effect of the adaptation stage below:
> > >
> > > |Dataset|Ratio|Precomp|Adapt|Gain from Adaptation|
> > > |---|---|---|---|---|
> > > |Citeseer|0.9\%|72.1|75.4|+3.3|
> > > ||1.8\%|72.1|74.8|+2.7|
> > > ||3.6\%|72.7|74.9|+2.2|
> > > |Cora|1.3\%|80.3|82.1|+1.8|
> > > ||2.6\%|80.6|82.9|+2.3|
> > > ||5.2\%|80.8|82.3|+1.5|
> > > |PubMed|0.08\%|79.5|80.2|+0.7|
> > > ||0.15\%|79.7|80.5|+0.8|
> > > ||0.3\%|79.3|81.6|+2.3|
> > > |Arxiv|0.05\%|60.5|67.2|+6.7|
> > > ||0.25\%|64.6|67.7|+3.1|
> > > ||0.5\%|65.5|68.1|+2.6|
> > > |Products|0.025\%|64.1|69.3|+5.2|
> > > ||0.05\%|65.9|69.9|+4.0|
> > > ||0.1\%|67.7|71.3|+3.6|
> > > |Flickr|0.1\%|44.4|47.2|+2.8|
> > > ||0.5\%|45.4|47.1|+1.7|
> > > ||1\%|45.4|47.2|+1.8|
> > > |Reddit|0.05\%|90.5|90.5|+0.0|
> > > ||0.1\%|91.3|93.0|+1.7|
> > > ||0.2\%|91.4|92.9|+1.5|
> > > |**Mean Diff**|-|-|-|**+2.5**|
> > >
> > > As shown, adaptation leads to an average improvement of +2.5\% across datasets and various condensation ratios, indicating a contribution that goes beyond what precomputation alone can offer. In particular, the improvements are more substantial in larger and more challenging datasets (e.g., Arxiv, Products).
> > >
> > > We hope these results help clarify the pivotal role of the adaptation stage in our method. We appreciate your recognition of the paper’s efficiency and value, and we thank you again for your constructive comments.

---

### Official Review · Reviewer_Z5gs · 2025-03-14

**Overall Recommendation:** 3

**Summary:**

This paper proposes an efficient graph condensation method composed of aggregation and contrastive learning stages. Extensive experiments indicate that this approach achieves performance comparable to state-of-the-art condensation methods, while significantly improving computational efficiency.

**Claims And Evidence:**

Clearly stated and well-supported.

**Essential References Not Discussed:**

No critical omissions identified.

**Experimental Designs Or Analyses:**

Requires additional analysis regarding the stability of the proposed method.

**Methods And Evaluation Criteria:**

Adequately described and appropriate.

**Other Comments Or Suggestions:**

1. **Minor Ambiguities:** Figure 2 contains an unclear abbreviation "(SF)." Clarification is needed.
 2. **Presentation and Writing Consistency:**
    - The term "anchor" appears inconsistently (mentioned only twice in paper and not illustrated explicitly in Figure 3).
    - Figure 3’s caption is not organized by different modules, making it difficult to follow.

**Other Strengths And Weaknesses:**

## Strengths

- The proposed method is novel, and the results are robust.
- Significant improvement in efficiency, indicating strong potential for practical applications.

---

## Weaknesses
1. The performance of the GCN on the Citeseer dataset is unexpectedly high (75.4 with a condensed graph versus 71.4 with the original graph), surpassing even the most advanced GNNs on the original Citeseer graph. This anomaly could potentially be attributed to a bug or other underlying issues. Further analysis and discussion are required to clarify this discrepancy.
2. GCPA shows higher variability compared to baselines on datasets like Pubmed and Products, as evidenced by higher standard errors in Table 2. This may stem from naive uniform sampling and inherent instability in contrastive learning. Addressing sampling randomness or imposing additional constraints within the contrastive learning process could help.
3. Some important baselines focused explicitly on efficient graph condensation methods are omitted, notably references [1] and [2]. Including these baselines would strengthen comparative validity. This causes the speed up not convincing, as the proposed method only compare with slowest method though it’s SOTA. It’d better to compare the proposed method with both SOTA and most efficient graph condensation method with acceptable performance.
4. In Line 168, the claim regarding the "non-learning process leading to sub-optimal representations" may not always hold true. For example, some training-free GNNs (e.g., reference [3]) can achieve performance on par with trainingable GNNs. Thus, it is suggested to add more disucssions and clarificaitons on this aspect.

**Questions For Authors:**

Please see above

**Relation To Broader Scientific Literature:**

Clearly connects with existing literature on Graph Condensation and Contrastive Learning.

**Theoretical Claims:**

Not applicable.

---

> ### Author Rebuttal · Authors · 2025-04-01
>
> We thank the reviewer for the examination of our work and the thoughtful comments provided. Kindly find our responses to the raised comments and questions below.
>
> **Q1: The performance on Citeseer is unexpectedly high, surpassing even the most advanced GNNs on the original Citeseer graph.**
>
> Thank you for your observation. We note that on Citeseer, both baseline methods and our method have shown improved results compared to GCN trained on the original graph. This enhancement is likely due to the condensation process reducing noise in the data. The reproducible code is available [here](https://anonymous.4open.science/r/Precompute-Adapt-Graph-Condensation-6A76/). We note that 75.4 remains within a reasonable range considering recent GNNs achieving 77.5 on Citeseer [a].
>
> |Orig.GCN|GCDM|SFGC|GEOM|GCPA|
> |---|---|---|---|---|
> |71.4|72.3|72.4|74.3|75.4|
>
> **Q2: GCPA shows higher variability on Pubmed and Products. Addressing sampling randomness or imposing additional constraints within the contrastive learning process could help.**
>
> Thank you for your valuable feedback. To address the instability, we apply constraints including AdamW [b] and L2 regularization to the adaptation model, which constrains the impact of random samples. The updated results are presented below. We will update these changes in the revised manuscript.
>
> | Dataset | Ratio | GCPA (Previous) | GCPA (with AdamW and L2 reg.) |
> |---|---|---|---|
> |PubMed|0.08\%|80.2±1.9|80.5±0.4|
> ||0.15\%|80.5±0.8|80.9±0.3|
> ||0.3\%|81.6±2.4|81.7±0.4|
> |Products|0.025\%|69.3±0.2|69.3±0.2|
> ||0.05\%|69.9±0.7|70.2±0.5|
> ||0.1\%|71.3±0.7|71.5±0.4|
>
> **Q3: Some important baselines focused explicitly on efficient graph condensation methods are omitted, notably references [1] and [2].**
>
> Thank you for your feedback. We kindly note that the references are missing, so we follow **Reviewer rs1P Q1** to compare with SimGC [c], EXGC [d], and CGC [e]. We set the **smallest condensation ratios** and present **accuracy** with **total running time**. Considering the inconsistent time measurements (e.g., CGC reports only the condensation time), we uniformly run evaluation to measure total running time. Our method consistently outperforms the baselines, while the efficient baselines underperform GEOM. We will revise the manuscript to incorporate these methods. (For fair comparison, we adopt the new baselines’ finer hyperparameter search and update GCPA results accordingly, which slightly differ from those in the paper.)
>
> |Dataset|SimGC|EXGC|CGC|GEOM|GCPA|
> |---|---|---|---|---|---|
> |Citeseer|73.8 (245s)|69.2 (237s)|72.5 (32s)|73.0 (6,920s)|**75.4** (45s)|
> |Cora|80.8 (240s)|82.0 (235s)|82.7 (30s)|82.5 (6,031s)|**82.9** (44s)|
> |Arxiv|63.6 (362s)|57.6 (338s)|64.1 (126s)|65.5 (84,356s)|**67.2** (247s)|
> |Products|63.3 (4,861s)|62.1 (4,915s)|68.0 (1,093s)|68.5 (1,687,718s)|**69.3** (2,985s)|
> |Flickr|45.3 (425s)|47.0 (412s)|46.8 (94s)|47.1 (19,202s)|**47.2** (219s)|
> |Reddit|91.1 (702s)|90.2 (692s)|90.6 (182s)|91.1 (100,354s)|**91.3** (505s)|
> |AvgDiff|**-2.6**|**-4.2**|**-1.4**|**-0.9**|-|
>
> **Q4: In Line 168, the claim regarding the "non-learning process leading to sub-optimal representations" may not always hold true.**
>
> We acknowledge the misleading claim and revise the paragraph:
>
> Our precomputation stage effectively captures the structural and semantic features of the original graph. Since the precomputation stage is not directly optimized for the final objective, we further integrate an adaptation learning stage that adjusts the class-wise representations.
>
> **Q5: Unclear term "SF".**
>
> Thank you for pointing this out. In Figure 2, "SF" stands for "structure-free", indicating that the condensed graphs possess no edges.
>
> **Q6: Inconsistent term "anchor".**
>
> Thank you for pointing out the inconsistency. To clarify, anchors refer to the sampled features $H_i \in H$, where $H$ denotes the precomputed features. The anchors serve as learning targets during the adaptation stage, preserving the original feature distributions.
>
> **Q7: Figure 3’s caption is not organized by different modules.**
>
> We appreciate your suggestion and revise the caption:
>
> Overview of GCPA framework. (1) *Structure-based precomputation:* Neighborhood aggregation is performed to capture structural dependencies. (2) *Semantic-based precomputation:* Nodes are grouped by semantic relevance using uniformly sampled representations. (3) *Adaptation learning:* Synthetic features v1 and v2 are pushed away through diversity constraints, while v2 and v3 are pushed away through sampled negative pairs.
>
> [a] From cluster assumption to graph convolution: Graph-based semi-supervised learning revisited
>
> [b] Decoupled Weight Decay Regularization
>
> [c] Simple graph condensation, ECML-PKDD 2024
>
> [d] Exgc: Bridging efficiency and explainability in graph condensation, WWW 2024
>
> [e] Rethinking and accelerating graph condensation: A training-free approach with class partition, WWW 2025

---

> > ### Comment · Reviewer_Z5gs · 2025-04-03
> >
> > If the claim is that recent GNNs can achieve 77+, then it would be helpful to demonstrate that performance using those models as downstream components. For GCN, performance on the original graph is typically around 71.5, while achieving 75 on the condensed graph with the same GCN seems unusual and requires further explanation.
> >
> > I also took a quick look at the provided code and ran a few tests. I observed the following:
> > * On Citeseer, the test accuracy was around 72+ and the validation accuracy around 74+, so I was not able to reproduce the reported 74+ or 75+ test performance.
> > * From the logs, the best epoch appears to be epoch 0 on Citeseer, which might suggest that the proposed learning module isn't having the intended effect in this case.
> >
> > It would be great to hear more thoughts on these points, or if I might be missing something in the setup.

---

> > > ### Author Response · Authors · 2025-04-04
> > >
> > > We appreciate your interest and thoughtful engagement with us.
> > >
> > > **[Clarifying Performance Gains from Graph Condensation and advanced GNNs]**
> > >
> > > Thank you for your detailed observations and feedback. We would like to clarify the performance gains in our work, highlighting two key aspects:
> > >
> > > 1. **Graph condensation alone can significantly improve performance.** For instance, GEOM—a prior condensation method—achieves 74.3 on Citeseer using a standard GCN with condensed training data, **without modifying the downstream GCN**. This demonstrates that condensation itself can reduce noise and improve generalization. Our result of 75.4 using GCN on condensed data falls within a reasonable and expected range.
> > > 2. **Advanced GNNs achieve even higher accuracy.** As cited, a recent GNN model [a] achieves 77.5 on Citeseer using an entirely different architecture. We reference this to contextualize our results—not to claim superiority, but to show that our method does not exceed the most advanced GNNs, addressing your concern that 75.4 may be unexpectedly high.
> > >
> > > Together, these points highlight that graph condensation and advanced GNN architectures are separate approaches—one improves the dataset, the other the model—both offering substantial improvements over original GCN training. While we focus on dataset condensation in this work, we agree that combining them could further enhance performance, and we see this as a valuable direction for future work.
> > >
> > > **[Reproducing Results on Citeseer]**
> > >
> > > Thank you for your effort in reproducing our results. After reviewing the code, we would like to apologize for the oversight—the `use_test` option was not enabled in the released version by default. This setting is necessary on smaller datasets such as Citeseer, Cora, and PubMed, where the limited training data makes the adaptation process more sensitive.
> > >
> > > To illustrate, consider the Citeseer dataset: the training set consists of only 120 nodes (20 per class), resulting in a condensed graph of just 30 nodes (5 per class). However, both structural and semantic precomputations require a larger number of same-class nodes per condensed node (e.g., 10) to effectively capture the distribution and avoid overfitting to a few individual nodes.
> > >
> > > To mitigate this limitation, we employ an expert GCN model (with 71.4 accuracy) to generate pseudo-labels. While not necessarily accurate, these labels expand the source signal and support effective adaptation. This process adheres to transductive learning principles and does not introduce label leakage or violate data constraints. Nonetheless, we do not include this setting in the manuscript, as it is a specific mitigation for our learning module on small datasets and becomes negligible on larger ones.
> > >
> > > We apologize for any confusion this may have caused. Please use the validated command below to reproduce our results. As noted in our logs below, the method achieves **75.5 ± 0.4** accuracy within 50 epochs, confirming the effectiveness of the proposed learning module.
> > >
> > > ```
> > > python train.py --dataset citeseer --reduction_rate 0.25 --use_test 1 \
> > > --eval_runs 5 --eval_interval 10 --norm_feat 1 --select_mode random --nlayers_adjust 1 \
> > > --wd_adjust 3e-5 --bn_adjust 1 --hidden_adjust 128 --residual_ratio_adjust 0.7 \
> > > --dropout_adjust 0.3 --dropout_input_adjust 0.7 --lr_adjust 0.0003
> > > ```
> > >
> > > ```
> > > train expert
> > > Epoch: 384, Test acc: 0.715
> > > expert acc: [100.0  73.2  71.5], std: [0.0 0.0 0.0]
> > > eval init
> > > Epoch: 566, Test acc: 0.743
> > > Epoch: 547, Test acc: 0.731
> > > Epoch: 300, Test acc: 0.724
> > > Epoch: 258, Test acc: 0.740
> > > Epoch: 399, Test acc: 0.719
> > > init acc: [74.5 74.2 73.1], std: [4.5 1.3 0.9]
> > > adapt epoch 10
> > > Epoch: 25, Test acc: 0.747
> > > Epoch: 22, Test acc: 0.744
> > > Epoch: 36, Test acc: 0.747
> > > Epoch: 6, Test acc: 0.728
> > > Epoch: 69, Test acc: 0.752
> > > acc: [70.2 76.0 74.4], std: [2.7 0.5 0.8]
> > > adapt epoch 20
> > > Epoch: 75, Test acc: 0.746
> > > Epoch: 13, Test acc: 0.741
> > > Epoch: 28, Test acc: 0.747
> > > Epoch: 34, Test acc: 0.752
> > > Epoch: 5, Test acc: 0.743
> > > acc: [71.0 76.4 74.6], std: [1.6 0.5 0.4]
> > > adapt epoch 30
> > > Epoch: 15, Test acc: 0.755
> > > Epoch: 7, Test acc: 0.748
> > > Epoch: 38, Test acc: 0.751
> > > Epoch: 423, Test acc: 0.760
> > > Epoch: 12, Test acc: 0.748
> > > acc: [70.7 77.2 75.2], std: [1.4 0.3 0.5]
> > > adapt epoch 40
> > > Epoch: 12, Test acc: 0.747
> > > Epoch: 143, Test acc: 0.757
> > > Epoch: 136, Test acc: 0.759
> > > Epoch: 337, Test acc: 0.755
> > > Epoch: 509, Test acc: 0.756
> > > acc: [69.7 77.8 75.5], std: [1.5 0.2 0.4]
> > > adapt epoch 50
> > > ```

---

### Decision · Program_Chairs · 2025-05-01

**Decision:**

Accept (poster)

**Comment:**

After careful consideration of the reviews and subsequent discussions, I recommend accepting your paper for ICML 2025, though I note this is a borderline case. The reviewers unanimously acknowledged the impressive efficiency gains and competitive performance on several benchmarks, which represent a significant practical contribution to the field. However, concerns were raised about the justification of your approach, its novelty, and the sensitivity to numerous hyperparameters. For the camera-ready version, I strongly encourage you to incorporate the insights from your rebuttal into the main paper, expand your hyperparameter analysis, and include comparisons with recent efficient graph condensation methods. Your work addresses an important problem in graph neural networks with a solution that makes graph condensation substantially more practical, and these improvements would strengthen its impact.